# VGGT-Motion: Motion-Aware Calibration-Free Monocular SLAM for Long-Range Consistency

Zhuang Xiong [* 1]  Chen Zhang [* 1]  Qingshan Xu [2]  Wenbing Tao [1]

## Abstract

Despite recent progress in calibration-free monocular SLAM via 3D vision foundation models, scale drift remains severe on long sequences. Motion-agnostic partitioning breaks contextual coherence and causes zero-motion drift, while conventional geometric alignment is computationally expensive. To address these issues, we propose VGGT-Motion, a calibration-free SLAM system for efficient and robust global consistency over kilometer-scale trajectories. Specifically, we first propose a motion-aware submap construction mechanism that uses optical flow to guide adaptive partitioning, prune static redundancy, and encapsulate turns for stable local geometry. We then design an anchor-driven direct Sim(3) registration strategy. By exploiting context-balanced anchors, it achieves search-free, pixel-wise dense alignment and efficient loop closure without costly feature matching. Finally, a lightweight submap-level pose graph optimization enforces global consistency with linear complexity, enabling scalable long-range operation. Experiments show that VGGT-Motion markedly improves trajectory accuracy and efficiency, achieving state-of-the-art performance in zero-shot, long-range calibration-free monocular SLAM.

## 1. Introduction

Monocular visual SLAM is a cornerstone of autonomous driving (Tateno et al., 2017; Milz et al., 2018; Cai et al., 2024; Su et al., 2025). While classical pipelines like ORB-

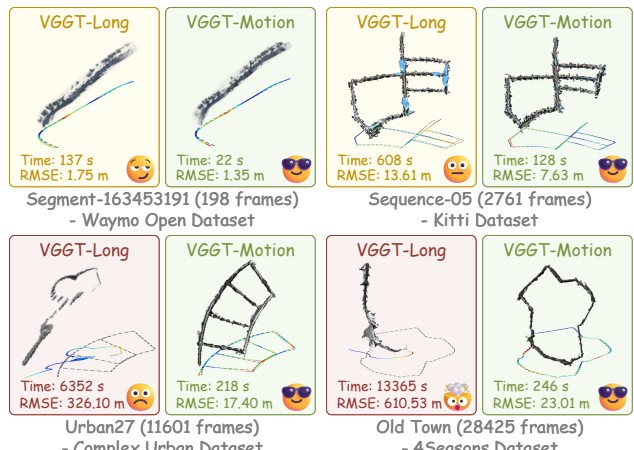

*Figure 1.* Comparison of the SOTA method VGGT-Long (Deng et al., 2025) and our VGGT-Motion across sequences with varying frame counts. While VGGT-Long suffers from significant drift as the number of frames increases, our VGGT-Motion maintains high accuracy and rapid inference speeds across all sequence durations.

SLAM series (Mur-Artal et al., 2015; Mur-Artal & Tardós, 2017) succeed in controlled settings, they often struggle in unstructured environments due to sensitivity to illumination and texture variations (Taketomi et al., 2017; Engel et al., 2017). Furthermore, inherent scale ambiguity and strict calibration requirements frequently lead to scale drift and geometric fragmentation, restricting their applicability to uncalibrated, in-the-wild video streams.

The emergence of 3D vision foundation models, such as DUSt3R (Wang et al., 2024), MASt3R (Leroy et al., 2024) and VGGT (Wang et al., 2025a), has introduced a transformative paradigm by jointly inferring camera pose, intrinsics, and dense geometry within unified differentiable frameworks. These models enable calibration-free reconstruction from raw image data, making SLAM in unconstrained environments a tangible possibility (Murai et al., 2025). However, the quadratic complexity ($O(N^2)$) of their Transformer-based architectures imposes severe memory constraints and computational bottlenecks, rendering direct deployment on long-horizon sequences infeasible.

To this end, some methods adopt divide-and-conquer strategies, such as sliding windows (Maggio et al., 2025) or sim-

---

[*]Equal contribution [1]National Key Laboratory of Science and Technology on Multi-spectral Information Processing, Huazhong University of Science and Technology, Wuhan, China [2]School of Information Science and Technology, University of Science and Technology of China, Hefei, China. Correspondence to: Qingshan Xu <qingshan.xu@ustc.edu.cn>, Wenbing Tao <wenbingtao@hust.edu.cn>.

*Proceedings of the $43^{rd}$ International Conference on Machine Learning*, Seoul, South Korea. PMLR 306, 2026. Copyright 2026 by the author(s).

ple chunking (Deng et al., 2025). As shown in Figure 1, while such rigid adaptations improve scalability, they introduce critical limitations that hinder efficiency and global consistency:

*1) Geometric Fragmentation in Motion-Agnostic Partitioning.* Existing monocular SLAM systems often rely on fixed-interval partitioning decoupled from camera dynamics. However, in the context of Transformer architectures, these rigid windowing strategies frequently fragment critical maneuvers, such as sharp turns, into disparate submaps, disrupting the global self-attention mechanism, which is essential for maintaining geometric consistency. While integrated sequences leverage a unified attention field for robust rotation and scale estimation, fragmented submaps lose this global geometric context, leading to inconsistent local trajectories and scale jumps at sequence boundaries.

*2) Spatio-temporal Redundancy and Hallucinated Zero-Motion Drift.* Dense temporal sampling in autonomous driving introduces two sources of redundancy for foundation-model-based SLAM. During static intervals such as traffic stops, transient sensor noise and subtle environmental dynamic factors dominate inter-frame differences and motion priors tend to "hallucinate" displacement, producing zero-motion drift. During dynamic motion, excessive frame rates lead to semantic feature saturation ($> 0.95$ similarity (Yuan et al., 2026)), yielding negligible geometric gain. These redundant streams further amplify the quadratic $O(N^2)$ cost of self-attention (Dosovitskiy et al., 2020), motivating adaptive filtering to retain informative frames while pruning redundancy.

*3) Contextual Asymmetry and Systematic Alignment Bias.* Existing submap alignment relies on overlap registration. However, we observe a systematic bias at submap boundaries: preceding submaps favor near-field density due to historical context, while succeeding submaps show a far-field predictive bias. This contextual asymmetry causes identical points to be reconstructed at divergent positions across frames, and ignoring it introduces systematic translation errors, limiting global consistency.

In this paper, we present VGGT-Motion, a calibration-free monocular SLAM system built on VGGT that efficiently maintains global consistency over long-range trajectories. The key idea is to exploit motion dynamics and structural regularities to enhance global consistency and computational efficiency. Specifically, we first introduce a motion-aware submap construction module that leverages optical flow to segment sequences into static, turning, and linear regimes. Parallax-based keyframe selection and redundancy pruning suppress zero-motion drift while maintaining an informative keyframe stream, and turning encapsulation preserves geometric stability during complex maneuvers. Next, to address alignment biases caused by contextual

asymmetry, we develop an anchor-driven direct $Sim(3)$ registration module. By leveraging context-balanced anchors as shared geometric pivots, we enable search-free, pixel-indexed dense alignment, bypassing costly feature matching and achieving linear $O(N)$ complexity. Finally, a lightweight pose graph optimization over sparse submaps enforces global consistency with minimal overhead.

Experiments on five diverse datasets show that our method achieves high-fidelity trajectory consistency and exceptional efficiency. On challenging zero-shot, long-sequence benchmarks (4Seasons (Wenzel et al., 2020), Complex Urban Dataset (Jeong et al., 2019), and A2D2 (Geyer et al., 2020)), it achieves an **84–96% reduction** in trajectory error and an **18–36× speedup** over the state-of-the-art (SOTA) VGGT-Long (Deng et al., 2025). It also remains robust to illumination changes, low-texture scenes, and high-speed motion.

In summary, our contributions are as follows:

- A motion-aware submap construction scheme adaptively partitions sequences based on camera dynamics, effectively suppressing zero-motion drift while preserving geometric integrity during complex maneuvers.
- Anchor-driven direct $Sim(3)$ registration resolves contextual asymmetry and enables search-free, dense geometric alignment with linear $O(N)$ complexity.
- A calibration-free SLAM framework delivers kilometer-scale global consistency with robust zero-shot generalization and high computational efficiency.

## 2. Related Work

**SfM & SLAM.** Classical 3D reconstruction largely builds on Structure-from-Motion (SfM) and Simultaneous Localization and Mapping (SLAM) (Hartley & Zisserman, 2003; Cadena et al., 2017). SfM pipelines (Schonberger & Frahm, 2016; Pan et al., 2024) achieve high reconstruction accuracy via global bundle adjustment (Triggs et al., 1999), whereas systems such as ORB-SLAM (Campos et al., 2021) perform incremental pose estimation with handcrafted features. Learning-based SLAM methods, including DROID-SLAM (Teed & Deng, 2021) and DPVO (Teed et al., 2023), enhance robustness through dense correspondence learning and differentiable matching within optimization-based pipelines. However, both classical and learned variants often assume known camera calibration, limiting their utility in unconstrained outdoor video processing (Taketomi et al., 2017).

**3D Foundation Models.** Unified 3D foundation models bypass epipolar search by directly regressing dense geometry and camera parameters. DUSt3R (Wang et al., 2024) enables calibration-free reconstruction via large-scale pretraining, MASt3R (Leroy et al., 2024) introduces 3D-grounded matching, and FAST3R (Yang et al., 2025) improves infer-

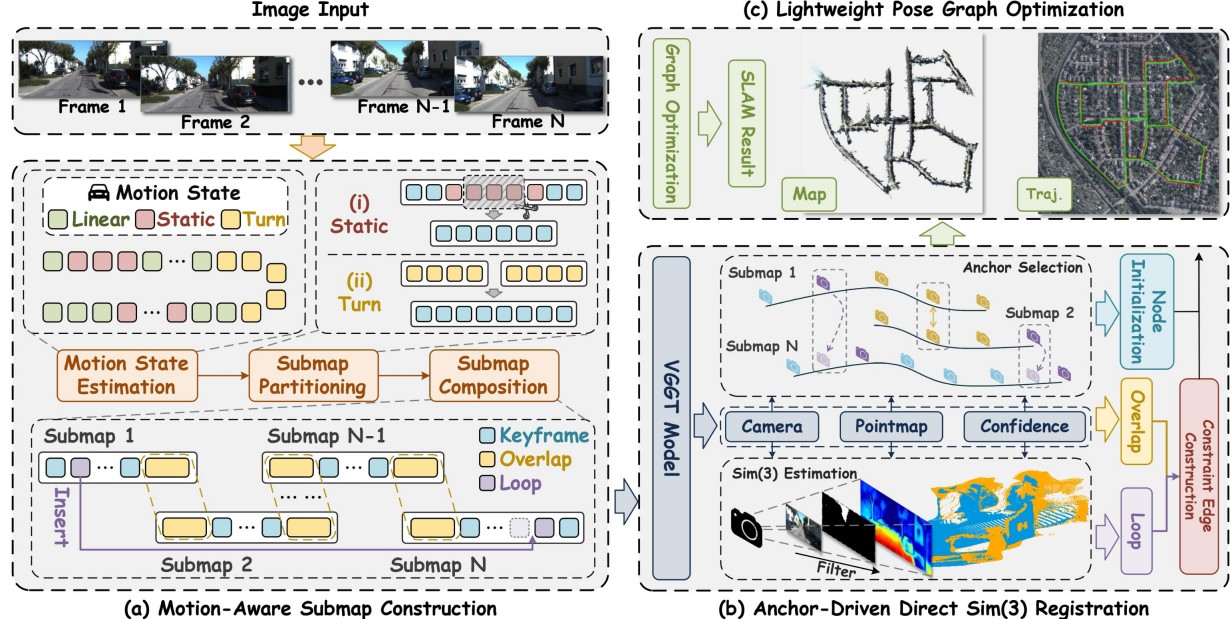

*Figure 2.* The pipeline of VGGT-Motion, consisting of three stages: (a) Motion-Aware Submap Construction, (b) Anchor-Driven Direct $Sim(3)$ Registration, and (c) Lightweight Pose Graph Optimization. During submap construction, estimated motion states are used to adaptively partition the sequence into base segments augmented with geometric anchors. For submap alignment, VGGT infers local dense geometry, and an anchor-driven strategy directly estimates $Sim(3)$ constraints. Finally, pose graph optimization recovers a globally consistent trajectory and map.

ence speed within the same batch paradigm. VGGT (Wang et al., 2025a) further integrates pose, intrinsics, and feature tracks within a Transformer architecture. Despite strong zero-shot generalization and local consistency, these models remain limited to short-sequence batch inference due to quadratic self-attention (Dosovitskiy et al., 2020; Deng et al., 2025), restricting scalability to long trajectories.

**Scalable Foundation-Based SLAM.** To scale foundation models to kilometer-level trajectories, existing efforts follow two directions. One line adopts streaming architectures with causal attention or incremental inference (Lan et al., 2025; Zhuo et al., 2025; Chen et al., 2025), extending temporal horizons but remaining susceptible to catastrophic forgetting (Kirkpatrick et al., 2017) and drift. The other line employs submap-based registration, as in VGGT-Long (Deng et al., 2025) and VGGT-SLAM (Maggio et al., 2025), which aligns overlapping chunks. However, motion-agnostic partitioning struggles with static redundancy and rapid turns, causing zero-motion drift and geometric fragmentation, and fixed-overlap alignment neglects contextual asymmetry at submap boundaries, leading to scale drift. These limitations call for a motion-aware calibration-free framework to maintain global consistency on long sequences.

# 3. Method

Given a monocular image sequence, our goal is to recover a globally consistent trajectory and dense geometry. We first construct motion-aware submaps that segment the sequence into reliable regimes and infer local geometry with the foundation model VGGT. Then, we develop an anchor-driven direct $Sim(3)$ registration algorithm to align submaps and optimize their poses. With the shared geometric anchors, we establish dense, search-free correspondences, achieving robust global consistency with linear computational complexity. Our pipeline is shown in Figure 2.

## 3.1. Motion-Aware Submap Construction

Naive temporal chunking in foundation-model SLAM triggers two systematic failures: (i) *zero-motion drift* during stop-and-go periods, where aligning noisy quasi-static predictions accumulates spurious motion; and (ii) *rotation-induced fragmentation*, where arbitrary cuts during turns disrupt attention context and scale consistency. As these errors originate from pre-inference grouping, they evade backend optimization. This motivates a motion-consistent construction scheme that classifies motion states via optical flow to adaptively select keyframes—pruning redundancy while preserving turning context.

**Motion State Estimation.** To quantify camera dynamics, we compute dense optical flow $\mathbf{F}_t : \Omega \rightarrow \mathbb{R}^2$ between consecutive frames $(I_{t-1}, I_t)$, where $\Omega$ denotes the image domain and $\mathbf{F}_t(\mathbf{u}) = (f_{x,t}(\mathbf{u}), f_{y,t}(\mathbf{u}))$. We derive two frame-wise scalar metrics: (i) a *static ratio* $r_{static}^{(t)}$, measur-

ing the proportion of quasi-static pixels:

$$r_{static}^{(t)} = \frac{1}{|\Omega|} \sum_{\mathbf{u} \in \Omega} \mathbb{I}\left(\|\mathbf{F}_t(\mathbf{u})\|_2 < \tau_{flow}\right), \qquad (1)$$

and (ii) a *turning score* $m_{turn}^{(t)}$, aggregating lateral flow magnitude:

$$m_{turn}^{(t)} = \frac{1}{|\Omega|} \sum_{\mathbf{u} \in \Omega} |f_{x,t}(\mathbf{u})|. \qquad (2)$$

After temporal Gaussian smoothing is applied to obtain the profiles $S_{static}(t)$ and $S_{turn}(t)$, a hierarchical classifier assigns each frame a discrete state $s(t) \in \{S, T, L\}$, representing *Static, Turning, and Linear* regimes:

$$s(t) = \begin{cases} S, & \text{if } S_{static}(t) > \tau_{static}, \\ T, & \text{else if } S_{turn}(t) > \tau_{turn}, \\ L, & \text{otherwise}, \end{cases} \qquad (3)$$

where $\tau_{static}$ and $\tau_{turn}$ are thresholds for detecting stationary and turning regimes.

**Motion-Aware Submap Partitioning.** Given the estimated motion state $s(t)$, we partition the input sequence into submaps that are geometrically informative and conducive to robust monocular scale estimation.

*Redundancy Filtering.* Foundation-model inference is highly sensitive to motion dynamics. First, in static intervals, sensor and environmental noise triggers "hallucinated movement" and zero-motion drift because the model fails to decouple noise from motion. Second, during motion, high semantic invariance masks informative geometric signals, necessitating a high–Signal-to-Noise Ratio (SNR) geometric flow for scale consistency. To mitigate these, we apply two complementary mechanisms:

- *Static Redundancy Pruning.* For static intervals, only the boundary frames (first and last) are retained. This suppresses zero-motion hallucination, reduces noise-driven drift, and significantly lowers computational overhead.
- *Parallax-Based Keyframe Selection.* Frames are admitted as keyframes only when the parallax relative to the previous keyframe exceeds $\tau_{palx}$, ensuring informative geometry and avoiding redundant processing.

*Topology-Aware Partitioning.* Turning trajectories are essential for monocular scale estimation due to their rich parallax structure. Arbitrarily fragmenting such segments disrupts geometric continuity and causes registration degradation at submap boundaries, especially when rotations dominate optical flow. To preserve structural coherence, we adopt:

- *Turning Segment Encapsulation.* Continuous turning intervals are encapsulated into a single submap to maintain parallax consistency and prevent scale discontinuities.

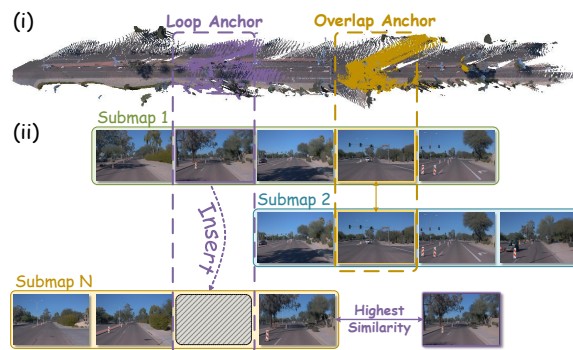

*Figure 3.* Context-balanced anchors. (i) Globally consistent reconstruction. (ii) Overlap and loop anchors for submap alignment.

- *Adaptive Linear Slicing.* For linear motion, a maximum frame budget $N_{max}$ controls submap length, balancing memory constraints with local geometric fidelity.

The resulting base segments $\mathcal{R}_k$ are further augmented with overlapping frames to enable seamless inter-submap alignment and loop closure. The detailed partitioning procedure is summarized in Algorithm 1 in the Appendix.

**Submap Composition with Geometric Anchors.** While motion-aware partitioning produces base segments $\mathcal{R}_k$, robust global alignment requires geometric overlap between submaps. To this end, each base segment is augmented with auxiliary anchors:

$$\mathcal{M}_k = \mathcal{R}_k \cup \mathcal{O}_k \cup \mathcal{L}_k. \qquad (4)$$

Here, $\mathcal{O}_k$ denotes the overlap keyframes ($|\mathcal{M}_k \cap \mathcal{M}_{k+1}| = N_{ovlp}$) ensuring stable adjacent alignment. Upon loop detection, $\mathcal{L}_k$ incorporates historical frames retrieved from past submaps via SALAD features (Izquierdo & Civera, 2024) and injected into the current queue. This corrects global drift without reprocessing past states, preserving linear complexity.

Each augmented submap sequence $\mathcal{M}_k$ is then fed to the VGGT model to produce a coherent geometric set:

$$\left\{(P_k^{(n)}, T_k^{(n)}, C_k^{(n)})\right\}_{n \in |\mathcal{M}_k|} = \Phi_{VGGT}(\mathcal{M}_k). \qquad (5)$$

Here, the outputs include local point maps $P_k$, camera poses $T_k$, and pixel-wise confidence maps $C_k$, all of which are used for subsequent inter-submap registration.

### 3.2. Anchor-Driven Direct Sim(3) Registration

To integrate submaps into a globally consistent trajectory, we estimate relative similarity transforms that rectify inter-submap gauge drift, especially scale ambiguity. Conventional descriptor matching (e.g., ORB (Rublee et al., 2011) or SIFT (Lowe, 2004)) is brittle in low-texture driving scenes and incurs quadratic $O(N^2)$ cost. In contrast, we

derive *dense, search-free* geometric correspondences from VGGT point maps to impose robust $Sim(3)$ constraints. This design yields $O(N)$ complexity (linear in valid pixels), enabling fast and reliable submap alignment.

**Context-Balanced Anchor Selection.** The Transformer-based architecture of VGGT conditions geometric predictions on temporal context. As a result, identical frames may yield inconsistent point maps when inferred under different temporal windows. To alleviate such context bias, we select reliable reference frames for alignment and categorize anchors into **overlap anchors** and **loop anchors**, as shown in Figure 3. For adjacent submaps, naive dense alignment over the overlap $\mathcal{W} = \mathcal{M}_i \cap \mathcal{M}_j$ suffers from non-linear scale drift at boundaries. Instead, we designate a compact temporal window surrounding the midpoint of $\mathcal{W}$ as the overlap anchor $\mathcal{I}_{ovlp} \triangleq \mathcal{N}(\text{mid}(\mathcal{W}))$. This selection ensures a balanced receptive field, minimizing boundary artifacts and maximizing geometric consistency between two adjacent submaps. For loop closure, context alignment is intrinsically established via our loop frame reuse mechanism. When a historical keyframe $h$ with the highest similarity is injected into the current sequence, it naturally serves as the loop anchor ($\mathcal{I}_{loop} \triangleq h$). Since $h$ retains its original metric state, no re-inference of past submaps is required, enabling direct loop closure alignment. In both scenarios, the shared anchor $\mathcal{I}_{a_{ij}}$ facilitates *direct, search-free* geometric correspondences. Since pixel coordinates lie on the same image lattice, we define the corresponding 3D points and confidence scores across distinct submap contexts as:

$$\mathbf{P}_i(\mathbf{u}) \triangleq P_i^{(a_{ij})}(\mathbf{u}), \qquad \mathbf{P}_j(\mathbf{u}) \triangleq P_j^{(a_{ij})}(\mathbf{u}),$$
$$\mathbf{C}_i(\mathbf{u}) \triangleq C_i^{(a_{ij})}(\mathbf{u}), \qquad \mathbf{C}_j(\mathbf{u}) \triangleq C_j^{(a_{ij})}(\mathbf{u}), \qquad (6)$$

where $\mathbf{u} \in \Omega$ indexes pixels, $\mathbf{P}_{\cdot}(\mathbf{u}) \in \mathbb{R}^3$ denotes the predicted 3D point in the respective submap coordinate system, and $\mathbf{C}_{\cdot}(\mathbf{u}) \in \mathbb{R}$ represents its confidence score.

**Pixel-Indexed Dense Correspondence.** Since both point maps $\mathbf{P}_i$ and $\mathbf{P}_j$ are inferred from the same anchor $\mathcal{I}_{a_{ij}}$, they share an inherent pixel-wise alignment despite the differing submap contexts. Consequently, each pixel index $\mathbf{u}$ establishes a deterministic geometric correspondence between the two submap coordinate systems:

$$\mathbf{x}_i^{\mathbf{u}} = \mathbf{P}_i(\mathbf{u}), \qquad \mathbf{x}_j^{\mathbf{u}} = \mathbf{P}_j(\mathbf{u}). \qquad (7)$$

We model their geometric relation by a similarity transform $\mathbf{S}_{ij} \in Sim(3)$:

$$\mathbf{x}_j^{\mathbf{u}} = \mathbf{S}_{ij}\,\mathbf{x}_i^{\mathbf{u}} + \epsilon_{\mathbf{u}}, \qquad (8)$$

where $\epsilon_{\mathbf{u}}$ accounts for prediction noise and residual context discrepancies. Crucially, this formulation yields $O(HW)$ dense constraints with trivial $O(1)$ association cost per pixel,

effectively circumventing descriptor extraction, nearest-neighbor search, and the quadratic complexity of conventional dense matching.

**Reliability-Aware Robust *Sim*(3) Estimation.** Since not all pixels provide reliable geometric constraints, we strictly filter the integration domain to exclude potential outliers. Specifically, we define the valid pixel set $\Omega_{valid}$ by intersecting high-confidence regions with semantic non-sky areas (obtained via a lightweight off-the-shelf sky segmenter (Liba et al., 2020)):

$$\Omega_{valid} \triangleq \big\{ \mathbf{u} \in \Omega \mid \min\big(\mathbf{C}_i(\mathbf{u}), \mathbf{C}_j(\mathbf{u})\big) > \tau_{conf} \big\}$$
$$\cap \{ \mathbf{u} \in \Omega \mid Mask_{sky}(\mathbf{u}) = 0 \}. \qquad (9)$$

We then estimate the optimal similarity transform $\hat{\mathbf{S}}_{ij} = \{s, \mathbf{R}, \mathbf{t}\}$ by robustly minimizing the alignment residual over these valid pixels:

$$\min_{s, \mathbf{R}, \mathbf{t}} \sum_{\mathbf{u} \in \Omega_{valid}} \rho_{Huber}\Big( \big\| s\mathbf{R}\mathbf{x}_i^{\mathbf{u}} + \mathbf{t} - \mathbf{x}_j^{\mathbf{u}} \big\|_2^2 \Big), \qquad (10)$$

where $\rho_{Huber}(\cdot)$ denotes the Huber loss function (Huber, 1992). Finally, we perform geometric verification using the inlier ratio $\eta_{ij}$ of the converged solution. If $\eta_{ij}$ falls below a threshold $\tau_{in}$, the constraint is discarded as unreliable; otherwise, the verified transformation is accepted and integrated into the global pose graph for trajectory refinement.

### 3.3. Lightweight Pose Graph Optimization

Given submap sequences $\{\mathcal{M}_k\}_{k=1}^K$, we construct a pose graph where nodes represent submap poses $\{\mathbf{X}_k\}_{k=1}^K$ with $\mathbf{X}_k \in Sim(3)$. Unlike frame-level methods, our optimization is performed at the submap level ($K \ll N$ nodes), significantly reducing computational complexity. Edges $\mathcal{E}$ incorporate relative $Sim(3)$ constraints $\hat{\mathbf{S}}_{ij}$ from adjacent overlaps and loop closures, each weighted by its inlier ratio $w_{ij} = \eta_{ij}$. This formulation bypasses expensive frame-level bundle adjustment by condensing dense geometric information into efficient submap-level constraints.

**Global Optimization.** For an edge $(i, j) \in \mathcal{E}$, the residual $\mathbf{r}_{ij}$ is defined in the Lie algebra via the logarithm map:

$$\mathbf{r}_{ij} = \log\Big(\hat{\mathbf{S}}_{ij}^{-1}\,\mathbf{X}_i^{-1}\mathbf{X}_j\Big) \in \mathbb{R}^7. \qquad (11)$$

We then estimate submap poses by minimizing the robustified sum of squared residuals:

$$\min_{\{\mathbf{X}_k\}} \sum_{(i,j) \in \mathcal{E}} \rho_{Huber}\big(\mathbf{r}_{ij}^{\top}\mathbf{\Omega}_{ij}\mathbf{r}_{ij}\big), \qquad (12)$$

where $\rho_{Huber}(\cdot)$ is the Huber loss and $\mathbf{\Omega}_{ij} = w_{ij}\mathbf{I}$ is the information matrix. We fix the gauge at $\mathbf{X}_1 = \mathbf{I}$ and solve Eq. 12 via Levenberg-Marquardt on Lie groups.

*Table 1.* Absolute Trajectory Error (ATE ↓) RMSE (m) comparison on the KITTI dataset. Following VGGT-Long, we additionally report Avg.* with the sequence 01 excluded. We further include VGGT-Long*, indicating our local re-evaluation under the same parameter settings as its paper. "TL" and "OOM" indicate Tracking Lost and Out-of-Memory errors respectively, while "LC" stands for Loop Closure. **Best** and second-best results are highlighted in **green** and light green , respectively.

| Method | LC | 00 | 01 | 02 | 03 | 04 | 05 | 06 | 07 | 08 | 09 | 10 | Avg. | Avg.* |
|---|---|---|---|---|---|---|---|---|---|---|---|---|---|---|
| *Detail* | | | | | | | | | | | | | | |
| *seq. frames* | - | 4541 | 1101 | 4661 | 801 | 271 | 2761 | 1101 | 1101 | 4071 | 1591 | 1201 | 2109 | 2210 |
| *seq. length (m)* | - | 2453.20 | 5067.23 | 3724.19 | 560.89 | 393.65 | 2205.58 | 1232.88 | 649.70 | 3222.80 | 1705.05 | 919.52 | 2012.24 | 1968.15 |
| *seq. speed (m/frame)* | - | 0.82 | 2.23 | 1.09 | 0.70 | 1.45 | 0.80 | 1.12 | 0.59 | 0.79 | 1.07 | 0.77 | 0.95 | 0.89 |
| *contains loop* | - | ✓ | × | × | × | × | ✓ | ✓ | ✓ | × | ✓ | × | - | - |
| *Calib.* | | | | | | | | | | | | | | |
| ORB-SLAM2 | × | 40.65 | 502.20 | 47.82 | **0.94** | 1.30 | 29.95 | 40.82 | 16.04 | 43.09 | 38.77 | **5.42** | 69.73 | 26.48 |
| ORB-SLAM2 | ✓ | 6.03 | 508.34 | **14.76** | 1.02 | 1.57 | **4.04** | **11.16** | 2.19 | 38.85 | 8.39 | 6.63 | 54.82 | **9.46** |
| DPVO | × | 113.21 | **12.69** | 123.40 | 2.09 | **0.68** | 58.96 | 54.78 | 19.26 | 115.90 | 75.10 | 13.63 | **53.61** | 57.70 |
| DROID-SLAM | - | 92.10 | 344.60 | 107.61 | 2.38 | 1.00 | 118.50 | 62.47 | 21.78 | 161.60 | 72.32 | 118.70 | 100.28 | 75.85 |
| *Uncalib.* | | | | | | | | | | | | | | |
| MASt3R-SLAM | ✓ | TL | TL | TL | TL | TL | TL | TL | TL | TL | TL | TL | - | - |
| CUT3R | × | OOM | OOM | OOM | 148.1 | 22.31 | OOM | OOM | OOM | OOM | OOM | OOM | - | - |
| Fast3R | × | OOM | OOM | OOM | OOM | OOM | OOM | OOM | OOM | OOM | OOM | OOM | - | - |
| VGGT | × | OOM | OOM | OOM | OOM | OOM | OOM | OOM | OOM | OOM | OOM | OOM | - | - |
| VGGT-SLAM | ✓ | 82.73 | 96.73 | 211.26 | 13.59 | 3.84 | 56.01 | 9.47 | 23.83 | 176.41 | 93.54 | 49.10 | 74.23 | 71.98 |
| VGGT-Long | ✓ | 8.67 | 121.2 | **32.08** | 6.12 | 4.23 | 8.31 | 5.34 | 4.63 | 53.10 | 41.99 | 18.37 | 27.64 | 18.28 |
| VGGT-Long* | ✓ | 9.37 | 93.30 | 60.14 | **5.87** | 4.05 | 13.61 | 6.01 | 4.04 | 60.64 | 31.72 | 23.44 | 28.38 | 21.89 |
| Ours | ✓ | **6.79** | **83.29** | 66.48 | 7.08 | **2.78** | **7.63** | **4.06** | **3.72** | **39.93** | **28.77** | 15.35 | **24.17** | **18.26** |

*Table 2.* Absolute Trajectory Error (ATE ↓) RMSE (m) comparison on Waymo Open dataset. Recent foundation-model-based methods are included. "TL" and "OOM" indicate Tracking Lost and Out-of-Memory errors, respectively. **Best** and second-best results are highlighted in **green** and light green , respectively.

| Segment ID | Calib. | 163453191 | 183829460 | 315615587 | 346181117 | 371159869 | 405841035 | 460417311 | 520018670 | 610454533 | Avg. |
|---|---|---|---|---|---|---|---|---|---|---|---|
| *Frame num.* | - | 198 | 199 | 199 | 199 | 196 | 199 | 198 | 199 | 198 | 198.0 |
| *Segment length* | - | 159.96 | 42.30 | 165.15 | 351.21 | 272.66 | 85.74 | 265.91 | 134.55 | 62.74 | 172.53 |
| *Segment speed* | - | 0.81 | 0.21 | 0.83 | 1.77 | 1.39 | 0.43 | 1.34 | 0.68 | 0.32 | 0.87 |
| *Traffic* | - | *Low* | *High* | *Low* | *Low* | *Medium* | *Low* | *Medium* | *Low* | *High* | - |
| DROID-SLAM | *Required* | 3.71 | 0.30 | 0.45 | 8.65 | 9.32 | 7.62 | 4.17 | TL | 0.26 | 4.40 |
| MASt3R-SLAM | *No Need* | 4.50 | 0.56 | 1.83 | 12.54 | 8.60 | 1.41 | 5.43 | 7.91 | 1.20 | 5.56 |
| CUT3R | *No Need* | 8.78 | 3.81 | 5.79 | 24.02 | 13.07 | 7.26 | 13.21 | 8.60 | 3.23 | 9.87 |
| Fast3R | *No Need* | OOM | OOM | OOM | OOM | OOM | OOM | OOM | OOM | OOM | - |
| VGGT | *No Need* | OOM | OOM | OOM | OOM | OOM | OOM | OOM | OOM | OOM | - |
| VGGT-SLAM | *No Need* | 1.82 | 0.64 | 1.42 | 12.02 | 7.44 | 2.14 | 2.95 | 7.99 | 0.57 | 4.11 |
| VGGT-Long | *No Need* | 1.75 | 2.63 | 0.56 | 3.45 | 3.34 | 1.44 | **1.54** | **2.55** | 0.46 | 2.00 |
| Ours | *No Need* | **1.35** | **0.14** | **0.40** | **3.10** | **2.00** | **1.08** | 2.16 | 3.93 | **0.12** | **1.59** |

**Trajectory Composition.** The global pose of any keyframe $I_n \in \mathcal{M}_k$ is obtained by composing the optimized submap pose with the local pose $T_k^{(n)}$:

$$\mathbf{T}_w^{(n)} = \mathbf{X}_k T_k^{(n)}, \quad n \in |\mathcal{M}_k|. \tag{13}$$

This yields the full global trajectory and a globally consistent reconstruction. Overall, our backend performs sparse $Sim(3)$ optimization over a small set of submaps, yielding a globally consistent trajectory at low computational cost.

## 4. Experiments

### 4.1. Experimental Setup

**Datasets.** To demonstrate robustness across diverse environments, we curate an evaluation suite comprising five outdoor datasets. Beyond the standard KITTI (Geiger et al., 2012) and Waymo Open (Sun et al., 2020) benchmarks,

we evaluate zero-shot performance on 4Seasons (Wenzel et al., 2020), Complex Urban (Jeong et al., 2019), and A2D2 (Geyer et al., 2020) to test generalization against varying weather, lighting, and motion patterns.

**Evaluation Metrics.** We report the Absolute Trajectory Error (ATE) after global $Sim(3)$ alignment to account for monocular scale ambiguity. Furthermore, on kilometer-scale trajectories, we report Translation Drift (%) to capture non-uniform scale drift and long-term structural inconsistencies that are often masked by global rigid alignment.

**Baselines.** We benchmark against a comprehensive set of state-of-the-art systems, including both foundation-model-based methods (VGGT-Long (Deng et al., 2025), VGGT-SLAM (Maggio et al., 2025), MASt3R-SLAM (Murai et al., 2025), Fast3R (Yang et al., 2025), CUT3R (Wang et al., 2025b)) and calibrated pipelines (ORB-SLAM2 (Mur-Artal & Tardós, 2017), DPVO (Teed et al., 2023), DROID-SLAM

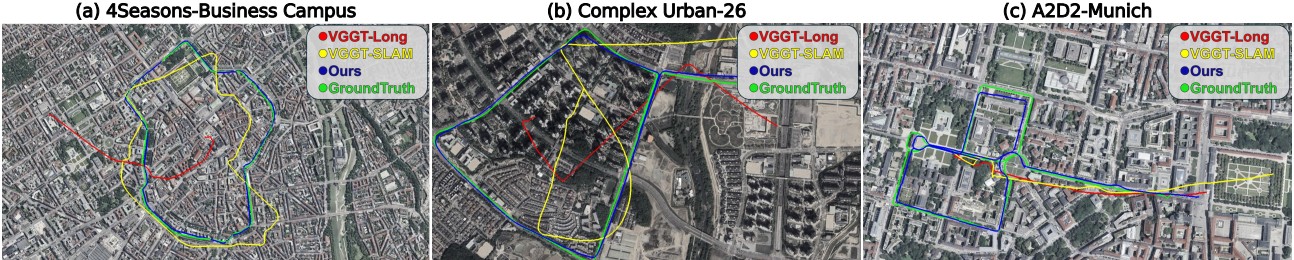

*Figure 4.* Qualitative results on generalization benchmarks. We visualize the estimated trajectories on (a) 4Seasons, (b) Complex Urban, and (c) A2D2 datasets. Our method exhibits long-range consistency across various challenging scenarios, whereas baseline methods fail.

*Table 3.* Experimental results on long-sequence generalization benchmarks. Absolute Trajectory Error (ATE) and Translation Drift are reported. "TL" and "OOM" denote Tracking Lost and Out-of-Memory failures, respectively.

| Method | Avg Frame: 16,495 4Seasons | | Avg Frame: 12,931 Complex Urban | | Avg Frame: 24,842 A2D2 | |
|---|---|---|---|---|---|---|
| | ATE (m) ↓ | Drift (%) ↓ | ATE (m) ↓ | Drift (%) ↓ | ATE (m) ↓ | Drift (%) ↓ |
| CUT3R | OOM | OOM | OOM | OOM | OOM | OOM |
| Fast3R | OOM | OOM | OOM | OOM | OOM | OOM |
| MASt3R-SLAM | TL | TL | TL | TL | TL | TL |
| VGGT-SLAM | 168.04 | 4.72% | 444.63 | 7.73% | 192.80 | 6.00% |
| VGGT-Long | 280.25 | 7.16% | 475.59 | 8.20% | 182.93 | 5.69% |
| **Ours** | **12.22** | **0.32%** | **35.48** | **0.58%** | **29.80** | **0.93%** |

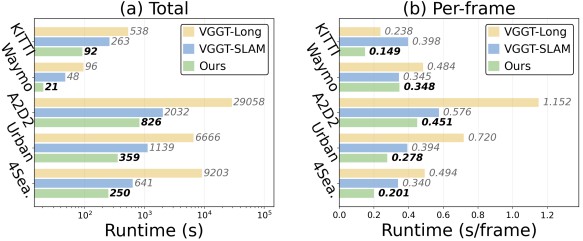

*Figure 5.* Runtime efficiency comparison on five different benchmarks. (a) Total runtime per scene (s) shown in log scale. (b) Average runtime per frame (s/frame).

([Teed & Deng, 2021](#)). For calibrated baselines, we supply ground-truth (GT) intrinsics to enforce an upper-bound comparison, while foundation-model-based systems are evaluated under their default uncalibrated settings.

**Implementation Details.** Experiments are performed on an NVIDIA RTX 3090 with an i7-13700KF CPU. Hyperparameters are fixed across benchmarks and dataset-specific thresholds are held constant for all sequences. Motion estimation uses $\{\tau_{flow}, \tau_{static}, \tau_{turn}\} = \{0.7, 0.6, 5\}$, and filtering/partitioning uses $\{\tau_{palx}, \tau_{conf}, N_{max}, N_{ovlp}\} = \{15, 0.5, 12, 5\}$. More details are provided in the Appendix.

### 4.2. Comparison on Standard Benchmarks

Following VGGT-Long, we evaluate on the KITTI and Waymo Open benchmarks, with results shown in Table 1 and Table 2. Foundation-model-based methods such as MASt3R-SLAM, CUT3R, and Fast3R frequently encounter Out-of-Memory (OOM) or Tracking-Lost (TL) failures, indicating limited scalability in large outdoor scenes. In contrast, VGGT-Motion and VGGT-Long scale reliably and competitive with some calibrated baselines that use GT intrinsics. Compared with VGGT-Long, our Motion-Aware Submap Construction mitigates zero-motion drift stemming from sensor noise and environmental disturbances, yielding more consistent trajectories that improve ATE on the KITTI dataset by **13%**. On highly dynamic Waymo segments, our context-balanced anchor selection and robust $Sim(3)$ estimation enforce geometric constraints under occlusions, yielding a further **20% improvement**. Visual results are provided in the Appendix.

### 4.3. Zero-Shot Generalization & Scalability

To evaluate the generalization capabilities of our method in unseen scenarios, we employ three large-scale datasets—4Seasons, Complex Urban, and A2D2. Crucially, none of these datasets were included in the pre-training corpus of the underlying VGGT model. These benchmarks present significantly greater challenges than standard KITTI or Waymo segments, featuring kilometer-scale trajectories with tens of thousands of frames that strictly test long-term stability against drift accumulation. They further exhibit drastic environmental shifts—such as seasonal appearance changes in 4Seasons and texture-poor urban canyons in Complex Urban, where traditional feature matching often fails. In addition, A2D2 introduces high-speed driving, which reduces effective frame overlap and poses higher challenges for registration stability.

Table 3 presents the quantitative results. The challenges of these datasets cause baselines like MASt3R-SLAM, Fast3R, and CUT3R to frequently suffer Out-of-Memory (OOM) or Tracking-Lost (TL) failures. While VGGT-Long and VGGT-SLAM are able to process longer sequences, they often exhibit severe scale drift, as illustrated in Figure 4. In contrast, our method delivers substantially more precise and robust trajectory estimation, with an **84–96% reduction** in both ATE and Drift over VGGT-Long. We attribute this success to our targeted algorithmic designs: Submap Composition with Geometric Anchors injects loop keyframes as geometric anchors to bridge appearance gaps across seasons; Anchor-Driven Direct $Sim(3)$ Registration estab-

*Table 4.* Ablation study on Redundancy Filtering on KITTI.

| Selection Strategy | ATE (m) ↓ | | ATE* (m) ↓ | |
|---|---|---|---|---|
| | Base | + Static Pruning | Base | + Static Pruning |
| Dense Sampling | 48.67 | **48.52** | 21.85 | **21.50** |
| Uniform Sampling | 34.14 | **34.07** | 16.09 | **15.95** |
| Covisibility-Based | 33.81 | **33.75** | 16.76 | **16.64** |
| **Parallax-Based** | 27.21 | **26.98** | 15.92 | **15.36** |

*Table 5.* Ablation of submap partitioning strategies on KITTI dataset. Topology-aware partitioning significantly outperforms heuristic baselines by preserving turning topology.

| Construction Strategy | Trigger Mechanism | ATE (m) ↓ | Drift (%) ↓ |
|---|---|---|---|
| Temporal Slicing | Temporal Length | 26.98 | 1.58 |
| Parallax-Triggered | Cumulative Parallax | 28.15 | 1.62 |
| **Topology-Aware** | **Turning Encapsulation** | **24.56** | **1.41** |

lishes dense, search-free geometric correspondences to ensure alignment in texture-poor urban canyons; and Motion-Aware Submap Construction improves long-term stability by mitigating zero-motion drift and rotation-induced fragmentation, which becomes critical under high-speed motion.

### 4.4. Efficiency

We evaluate computational efficiency by measuring the average end-to-end runtime under identical hardware configurations, excluding disk I/O. As shown in Figure 5, our pipeline consistently outperforms state-of-the-art foundation-model-based methods (VGGT-Long and VGGT-SLAM) across all five benchmarks. The performance gap is particularly large on long-sequence datasets (4Seasons, Complex Urban, and A2D2), where we achieve an **18–36× speedup** in total processing time over VGGT-Long. We further observe substantial reductions in per-frame latency. These gains stem primarily from our Redundancy Filtering and Direct $Sim(3)$ Registration. By actively filtering redundant frames, we minimize unnecessary inference as the sequence scales, while enabling rapid submap alignment.

### 4.5. Ablation Studies

We conduct ablation studies on KITTI to assess the contribution of each component in our pipeline. Our analysis focuses on how Motion-Aware Submap Partitioning (MASP) and Anchor-Driven Direct $Sim(3)$ Registration (ADSR) alleviate intrinsic challenges in foundation-model-based SLAM. To isolate effects, we adopt an incremental ablation protocol in which modules are added in pipeline order, with each stage initialized from the best configuration of the previous step. The baseline employs dense keyframe sampling, fixed temporal partitioning, and minimal inter-submap overlap.

**Redundancy Filtering.** First, We first evaluate the filtering mechanism in MASP, which synergizes Parallax-based Keyframe Selection with Static Redundancy Pruning. Re-

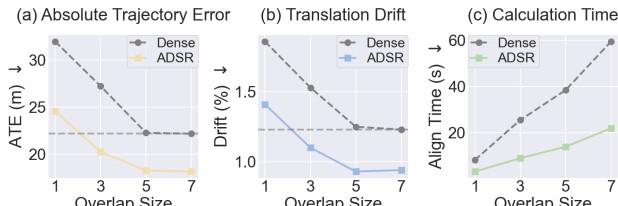

*Figure 6.* Impact of overlap size and submap alignment strategy.

sults are reported in Table 4 using the overall mean ATE (excluding Seq. 01) and a specialized metric, ATE*, specifically designed for stop-and-go sequences (00, 05, 07, 08). Observations reveal that naive dense (using all frames) or uniform sampling paradoxically degrades accuracy. Instead of providing valid geometric constraints, these redundant frames accumulate sensor noise and environmental disturbances. Even covisibility-based selection, which triggers keyframe insertion based on feature overlap, reduces redundancy but prioritizes visual continuity over spatial displacement. Consequently, it often yields narrow baselines, resulting in ill-conditioned geometric constraints. Conversely, our Parallax-based Selection yields the most significant gains. By strictly admitting frames based on geometric displacement, it ensures high-SNR constraints and maximizes information density while rejecting ill-conditioned observations. Complementarily, Static Redundancy Pruning enhances performance across all baselines. By eliminating the primary source of noise-induced drift during stops, it simultaneously improves accuracy and reduces computational overhead.

**Submap Partitioning.** Building upon the optimal redundancy filtering strategy established above, we next evaluate different submap partitioning mechanisms in Table 5. Standard temporal slicing imposes arbitrary boundaries that often sever continuous geometric structures. Surprisingly, parallax-triggered partitioning performs even worse. Since parallax accumulates most rapidly during turns, threshold-based triggers tend to inadvertently fragment high-curvature segments—the very regions critical for monocular scale observability. Our Topology-Aware strategy in MASP prevents this failure mode via Turning Segment Encapsulation. By preserving turning maneuvers as indivisible submaps, we maintain structural integrity and prevent scale fractures at submap boundaries.

**Submap Alignment.** Finally, we evaluate the impact of submap overlap size and alignment strategy. Given the strong SOTA performance of VGGT-Long in large-scale outdoor scenes, we adopt its submap alignment scheme as the baseline. The baseline performs dense frame-wise registration over the entire overlap region, indiscriminately aggregating noisy and redundant observations. In contrast, our ADSR adopts a select-and-filter strategy. We select the temporal central region of the overlap as the anchor to

maximize geometric coverage while mitigating boundary-induced context bias, and retain only points corresponding to high-confidence pixels within this anchor. This design distills the registration task into a compact, high-SNR region optimization problem. As a result, ADSR achieves more robust registration at substantially lower computational cost, avoiding the redundancy and noise accumulation inherent to dense alignment. As shown in Figure 6, our submap alignment method consistently outperforms the baseline method in both performance and efficiency.

## 5. Conclusion

We present a robust, calibration-free monocular SLAM framework that effectively addresses the scalability and drift bottlenecks inherent in foundation-model-based approaches. Central to our approach is a motion-aware data organization strategy that adaptively partitions sequences based on camera dynamics. This design actively prunes static redundancy to eliminate zero-motion drift, while encapsulating complex maneuvers to preserve local geometric integrity. Furthermore, our anchor-driven direct $Sim(3)$ registration exploits context-balanced anchors and intrinsic pixel-indexing to enable search-free dense alignment. Together, these designs achieve superior zero-shot generalization and low computational overhead on kilometer-scale datasets. We hope this work offers valuable insights into addressing efficiency and consistency challenges, advancing foundation-model-based monocular SLAM toward real-world deployment.

## Acknowledgements

This work was supported by the National Natural Science Foundation of China under Grants 6257074579, and the Hubei Provincial Science and Technology Plan Project under Grant No. 2025BEB006.

## Impact Statement

This paper presents work whose goal is to advance the field of Machine Learning. There are many potential societal consequences of our work, none which we feel must be specifically highlighted here.

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

# A. Motion-Aware Submap Partitioning Algorithm

To stabilize optimization and reduce redundancy, we propose a motion-aware partitioning strategy that leverages camera dynamics (static/linear/turning). The detailed procedure is summarized in Algorithm 1.

**Algorithm Details.**    The algorithm outlines the **Motion-Aware Submap Partitioning** procedure. It executes in two stages to generate base submaps from the raw stream. In the first stage (**Redundancy Filtering**), we decouple sensor noise and environmental disturbances from dynamics. For **static intervals** ($s(t) = $ S), we perform specific pruning where *only the boundary frames* are retained to suppress zero-motion drift. For **dynamic intervals**, we employ a parallax-based selector that admits a frame only if its parallax exceeds $\tau_{\mathrm{palx}}$, ensuring informative geometry. In the second stage (**Topology-Aware Partitioning**), we segment the selected keyframes. **Turning motions** ($s(t) = $ T) are encapsulated as atomic units without splitting to preserve local consistency. **Linear motions** ($s(t) = $ L) are adaptively sliced based on a maximum budget $N_{\mathrm{max}}$, producing the final set of base submaps $\{\mathcal{R}_k\}$ for downstream composition.

---

**Algorithm 1** Motion-Aware Submap Partitioning

---

1: **Input:** image sequence $\mathcal{I} = \{I_t\}_{t=1}^T$; motion state sequence $\mathcal{S} = \{s(t)\}_{t=1}^T$
2: **Hyper-parameters:** parallax threshold $\tau_{\mathrm{palx}}$; max keyframes per submap $N_{\mathrm{max}}$; static boundary window $\omega$
3: **Output:** base submaps $\{\mathcal{R}_k\}_{k=1}^K$, each $\mathcal{R}_k \subseteq \mathcal{I}$
4: {**Definitions (used in the algorithm)**}
5: $s(t) \in \{\mathrm{S}, \mathrm{L}, \mathrm{T}\}$ denotes **static / linear / turning** motion state at time $t$
6: $\mathrm{Palx}(I_a, I_b)$ is a scalar parallax proxy (larger $\Rightarrow$ more viewpoint change)
7: $\mathrm{IsStaticBoundary}(t, \mathcal{S}, \omega)$ returns True if $s(t) = $ S and $t$ is within $\omega$ frames of a non-static state transition
8:     (i.e., near S $\leftrightarrow$ (L or T) boundaries); used to preserve critical constraints around motion changes.
9: {**Initialization**}
10: $k \leftarrow 1$; $\mathcal{R}_1 \leftarrow \emptyset$; $t_{\mathrm{last}} \leftarrow 1$; $I_{\mathrm{last}} \leftarrow I_1$; $s_{\mathrm{prev}} \leftarrow s(1)$
11: **for** $t = 1$ **to** $T$ **do**
12:     {**Step 1: Redundancy Filtering (Keyframe Selection)**}
13:     $\delta_t \leftarrow$ **False** {$\delta_t$ indicates whether $I_t$ is selected into any submap}
14:     **if** $s(t) = $ S **then**
15:         {In static intervals, most frames are redundant. Keep only boundary frames for anchoring.}
16:         **if** $\mathrm{IsStaticBoundary}(t, \mathcal{S}, \omega)$ **then**
17:             $\delta_t \leftarrow$ **True** {retain boundary frames in static segments}
18:         **end if**
19:     **else**
20:         {In dynamic intervals (linear/turning), select by sufficient viewpoint change.}
21:         **if** $\mathrm{Palx}(I_t, I_{\mathrm{last}}) > \tau_{\mathrm{palx}}$ **then**
22:             $\delta_t \leftarrow$ **True**
23:             $I_{\mathrm{last}} \leftarrow I_t$; $t_{\mathrm{last}} \leftarrow t$
24:         **end if**
25:     **end if**
26:     {**Step 2: Topology-Aware Partitioning (Submap Slicing)**}
27:     **if** $\delta_t = $ **True then**
28:         **if** $s(t) = $ T **then**
29:             {Turning encapsulation: avoid cutting submaps during turns to prevent inconsistent local geometry.}
30:             $\mathcal{R}_k \leftarrow \mathcal{R}_k \cup \{I_t\}$
31:         **else**
32:             {Linear motion: adaptively slice submaps to control drift and keep local BA stable.}
33:             **if** $|\mathcal{R}_k| \geq N_{\mathrm{max}}$ **or** $s_{\mathrm{prev}} = $ T **then**
34:                 {Start a new submap if: (i) current submap is too large, or (ii) we just exited a turning interval.}
35:                 **Finalize** $\mathcal{R}_k$ {store $\mathcal{R}_k$ into output list}
36:                 $k \leftarrow k + 1$
37:                 $\mathcal{R}_k \leftarrow \emptyset$
38:             **end if**
39:             $\mathcal{R}_k \leftarrow \mathcal{R}_k \cup \{I_t\}$
40:         **end if**
41:     **end if**
42:     $s_{\mathrm{prev}} \leftarrow s(t)$
43: **end for**
44: {**Finalize the last submap (if non-empty)**}
45: **if** $|\mathcal{R}_k| > 0$ **then**
46:     **Finalize** $\mathcal{R}_k$
47: **end if**

---

# B. Discussion & Limitation

## B.1. Ablation Study

### B.1.1. LOOP KEYFRAME REUSE

Table 6 evaluates keyframe reuse directionality on loop closure. Regarding reuse direction of loop anchors, unidirectional reuse consistently outperforms bidirectional reuse. While bidirectional injection offers more constraints, it paradoxically degrades performance due to stability asymmetry: the historical submap is established, whereas current observations often contain transient noise. Injecting current frames back into history effectively corrupts the clean context. In contrast, unidirectional reuse strictly anchors the drifting current state to the frozen history, preventing noise propagation.

*Table 6.* Ablation of loop closure strategies on KITTI. Unidirectional loop keyframe reuse within CBA yields the best performance by preventing historical context pollution.

| Loop Closure Strategy | Loop Anchor Reuse Mode | ATE (m) ↓ | Drift (%) ↓ |
|---|---|---|---|
| CBA loop anchors | Bidirectional | 18.31 | 0.930 |
| **CBA loop anchors** | **Unidirectional** | **18.26** | **0.928** |

### B.1.2. HYPERPARAMETER SENSITIVITY

To evaluate the robustness of our motion-aware partitioning strategy, we conduct a sensitivity analysis over the core hyperparameters on KITTI and Waymo. All other parameters are fixed when varying a single hyperparameter. As shown in Table 7, the performance remains stable under a broad range of settings. The default configuration, highlighted in gray, provides a balanced trade-off across datasets and is used consistently unless otherwise specified.

### B.1.3. RUNTIME AND EFFICIENCY

We further compare the runtime efficiency with DROID-SLAM, a representative calibrated dense SLAM system. All runtimes are reported in seconds per frame on the same hardware platform. As shown in Table 8, VGGT-Motion achieves comparable runtime on short Waymo segments and is slightly faster on medium-length KITTI sequences. This is because the proposed motion-aware pruning strategy reduces redundant inference windows and limits the context size processed by the foundation model.

In addition, the optical-flow-based motion classification introduces only marginal overhead. We use the classical Farneback optical flow implemented in OpenCV, which takes only 3.3 ms per frame on average, with 95% of frames processed within 10 ms. The additional GPU memory consumption is approximately 522 MB, which is small compared with the memory footprint of foundation-model inference. Therefore, the motion-aware front-end improves long-sequence scalability without becoming a runtime or memory bottleneck.

## B.2. Discussion

The results presented in this work highlight a fundamental shift in monocular SLAM. Rather than relying solely on increasingly complex optimization backends, global consistency can be achieved by explicitly respecting motion dynamics and leveraging strong structural priors provided by modern foundation models. This perspective suggests that robustness at scale is not only an optimization problem, but also a representation and motion-awareness problem.

**Model-Agnostic Modularity and Metric Flexibility.** The proposed framework is intentionally designed to be modular and model-agnostic. While our experiments are conducted using the VGGT (Wang et al., 2025a) foundation model, the core components of the system—including motion-aware submap partitioning and inter-submap alignment—are independent of any specific geometric predictor. As demonstrated in Figure 7, the framework can readily incorporate other emerging 3D vision foundation models, such as Depth Anything series (Yang et al., 2024; Lin et al., 2025), Pi3 (Wang et al., 2025c), or Mapanything (Keetha et al., 2025), without architectural changes.

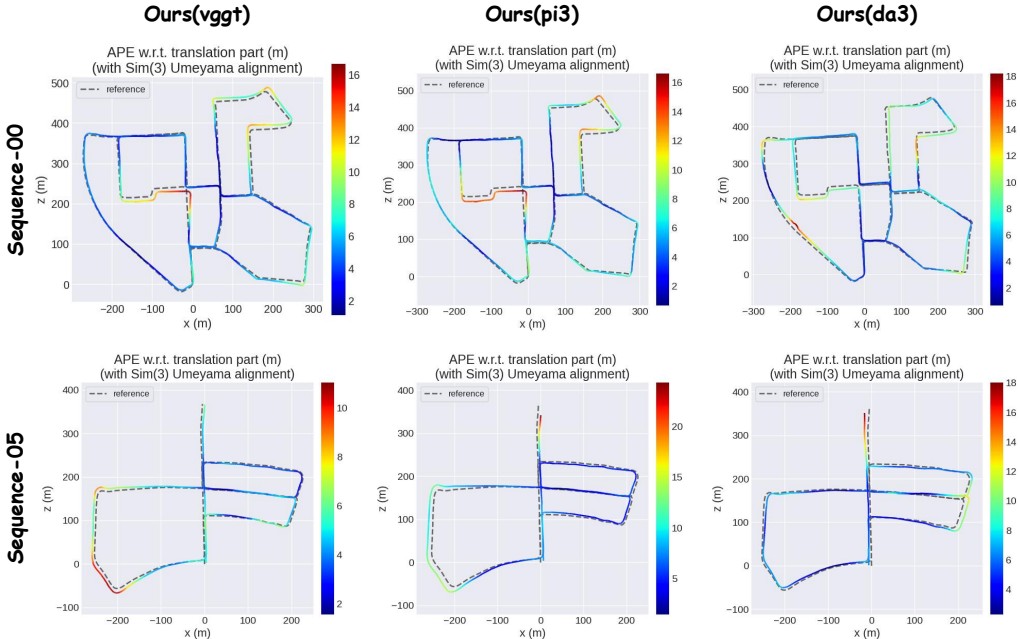

*Figure 7.* KITTI dataset sequence 00 & sequence 05 trajectory visualizations on different foundation models.

Importantly, this modularity enables flexible choices of geometric constraints based on the model's capabilities. For foundation models providing only relative scale, the system operates in a similarity space using $Sim(3)$ alignment. Conversely, models that infer near-metric geometry through multi-modal pre-training (eg. Mapanything) allow the pipeline to operate directly in $SE(3)$. As shown in Figure 8 (KITTI visualization), fixing the scale factor and reducing degrees of freedom during registration significantly improves trajectory stability and avoids monocular scale ambiguities.

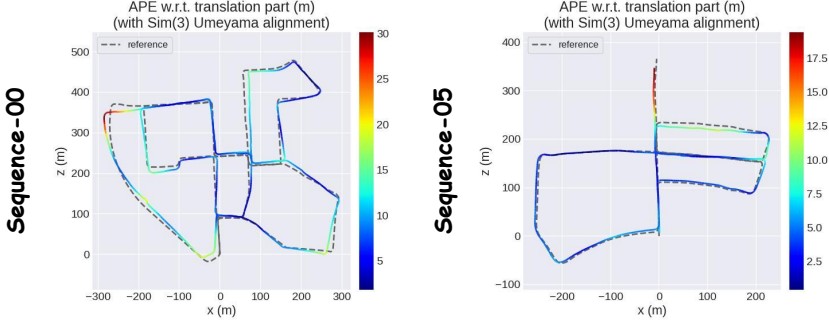

*Figure 8.* KITTI dataset sequence 00 & sequence 05 trajectory visualizations on DepthAnything 3(DA3) with *SE(3)* alignment.

**Coupling Between Geometric Priors and SLAM Performance.** Our study also reveals that overall trajectory fidelity is intrinsically coupled with the geometric inference quality of the underlying foundation model. This observation suggests a favorable property: as foundation models continue to improve through large-scale pre-training, the SLAM system can immediately benefit without additional retraining or algorithmic redesign. This coupling is quantitatively validated in Table 9, where switching to more advanced predictors like DepthAnything 3(DA3) (da3) consistently reduces ATE RMSE across Waymo segments. The corresponding trajectory visualizations in Figure 9 (Waymo visualization) qualitatively confirm how improved geometric priors translate directly into higher global consistency.

*Table 9.* Quantitative Results on Waymo Dataset (ATE RMSE [m]).

| Segment ID | 163453191 | 183829460 | 315615587 | 346181117 | 371159869 | 405841035 | 460417311 | 520018670 | 610454533 | Avg. |
|---|---|---|---|---|---|---|---|---|---|---|
| Ours (vggt) | 1.3459 | 0.1391 | 0.4042 | 3.0962 | 2.0029 | 1.0814 | 2.1597 | 3.9330 | 0.1213 | 1.5871 |
| Ours (pi3) | **1.2991** | 0.1744 | **0.3100** | 1.6821 | 2.5085 | **0.6771** | **1.0645** | 5.6316 | **0.0982** | 1.4939 |
| Ours (da3) | 1.7731 | **0.1239** | 1.4121 | **1.2567** | 2.0014 | 0.8831 | 1.2450 | **1.3046** | 0.1490 | **1.1277** |

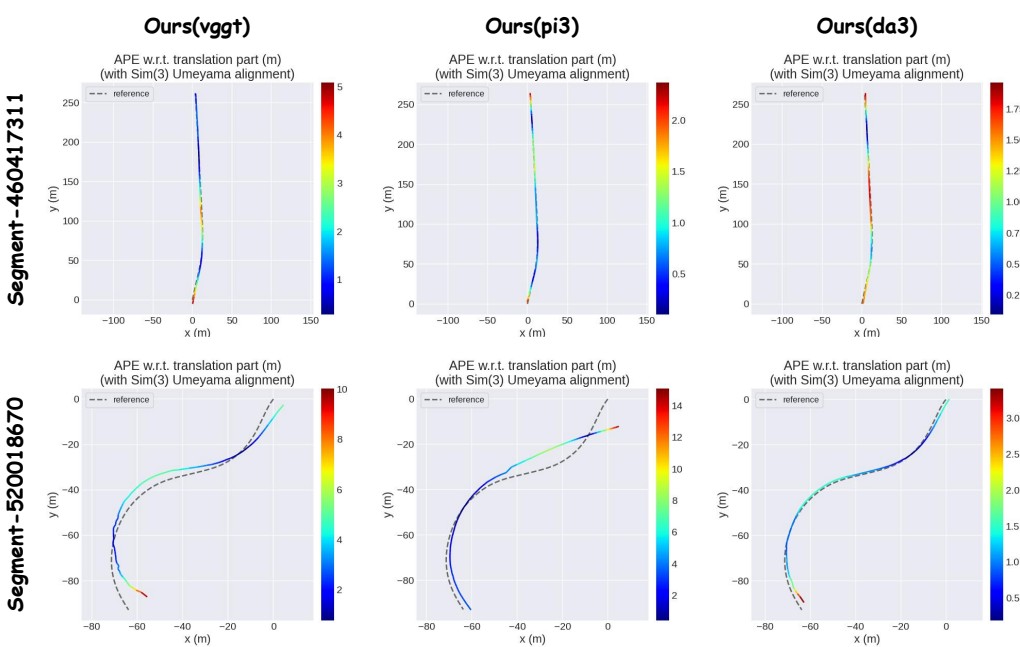

*Figure 9.* Waymo Open dataset Segment-460417311 & Segment-520018670 trajectory visualizations on different foundation models.

**Evolution Toward Renderable Scene Representations.** Beyond point-cloud-based mapping, the proposed pipeline opens new opportunities for integrating feed-forward models that predict explicit, renderable scene representations. Recent models such as Anysplat (Jiang et al., 2025) and DepthAnything 3(DA3) enable direct prediction of 3D Gaussian Splatting (3DGS) parameters from monocular inputs. By replacing intermediate point clouds with such representations, our framework can generate high-fidelity maps in a fully feed-forward manner. As illustrated in Figure 10, this evolution supports real-time novel-view synthesis and richer downstream applications in robotics and embodied AI, moving beyond traditional sparse or dense point cloud maps.

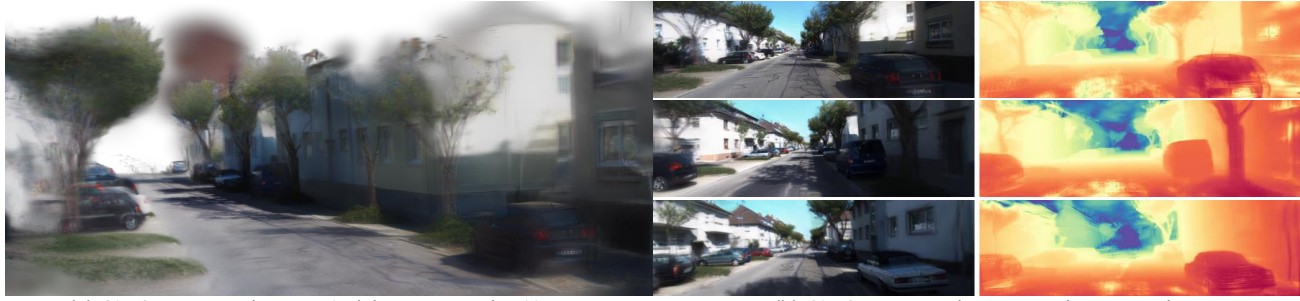

**(a) 3D Gaussian Splatting Model in SuperSplat Viewer**      **(b) 3D Gaussian Splatting rendering result**

*Figure 10.* Visualization of the feed-forward 3D Gaussian Splatting scene. (a) Feed-forward Gaussian Model predicted by DepthAnything 3(DA3), visualized in SuperSplat. (b) Rendering results from the 3DGS model: the left side shows the rendered RGB images, and the right side shows the corresponding rendered depth maps

### B.3. Limitation

Despite the promising results, our system has several limitations that invite future research.

**First**, although our framework is model-agnostic, the offline end-to-end runtime is inevitably bottlenecked by the inference latency of the underlying foundation model. While we mitigate this via keyframe selection, deploying the full pipeline on resource-constrained edge devices (e.g., micro-drones) remains a challenge without model quantization or distillation.

**Second**, while our motion-aware partitioning effectively handles camera dynamics (static, linear, turning), the system currently assumes a mostly rigid scene. In highly dynamic environments crowded with moving objects (e.g., busy intersections), the geometric priors from the foundation model might be contaminated, potentially affecting the mapping consistency.

**Third**, similar to traditional monocular SLAM, our system faces accumulated **scale, rotation, and translation drift**, particularly during **extended intervals between loop closures**. Although integrating foundation models that infer near-metric geometry (e.g., Mapanything, DepthAnything 3(DA3)) can mitigate scale ambiguity and enhance local consistency, relying solely on learned visual priors is often insufficient to guarantee bounded global error. Consequently, achieving high-precision metric accuracy in large-scale environments still necessitates the fusion of auxiliary sensors (e.g., IMU) to provide robust absolute constraints.

*Table 7.* Hyperparameter sensitivity analysis on KITTI and Waymo. Results are ATE RMSE (m). The default setting is highlighted in gray.

| KITTI | | | | | | | | | | | | |
|---|---|---|---|---|---|---|---|---|---|---|---|---|
| **Setting** | 00 | 01 | 02 | 03 | 04 | 05 | 06 | 07 | 08 | 09 | 10 | Avg. |
| $\tau_{\text{flow}} = 0.4$ | 6.800 | 83.293 | 66.482 | 7.084 | 2.776 | 7.630 | 4.059 | 3.838 | 39.984 | 28.773 | 15.497 | 24.201 |
| $\tau_{\text{flow}} = 0.7$ | 6.788 | 83.293 | 66.482 | 7.084 | 2.776 | 7.630 | 4.059 | 3.724 | 39.930 | 28.773 | 15.354 | 24.172 |
| $\tau_{\text{flow}} = 1.0$ | 6.824 | 83.293 | 66.482 | 7.084 | 2.776 | 7.665 | 4.059 | 3.748 | 39.936 | 28.773 | 15.325 | 24.179 |
| $r_{\text{turn}} = 3$ | 5.612 | 93.599 | 54.202 | 8.112 | 4.623 | 7.920 | 8.338 | 3.502 | 35.682 | 36.760 | 10.902 | 24.478 |
| $r_{\text{turn}} = 5$ | 6.788 | 83.293 | 66.482 | 7.084 | 2.776 | 7.630 | 4.059 | 3.724 | 39.930 | 28.773 | 15.354 | 24.172 |
| $r_{\text{turn}} = 7$ | 5.543 | 85.722 | 74.512 | 7.016 | 2.776 | 7.109 | 5.099 | 5.847 | 34.035 | 31.772 | 15.402 | 24.985 |
| $r_{\text{static}} = 0.5$ | 6.774 | 83.293 | 66.295 | 7.084 | 2.776 | 7.559 | 4.059 | 3.721 | 39.930 | 28.773 | 15.353 | 24.147 |
| $r_{\text{static}} = 0.6$ | 6.788 | 83.293 | 66.482 | 7.084 | 2.776 | 7.630 | 4.059 | 3.724 | 39.930 | 28.773 | 15.354 | 24.172 |
| $r_{\text{static}} = 0.7$ | 6.788 | 83.293 | 66.482 | 7.084 | 2.776 | 7.660 | 4.059 | 3.838 | 39.984 | 28.773 | 15.431 | 24.197 |
| $N_{\max} = 8$ | 9.587 | 118.250 | 72.944 | 13.586 | 2.673 | 8.246 | 3.946 | 4.100 | 35.240 | 35.571 | 24.753 | 29.900 |
| $N_{\max} = 12$ | 6.788 | 83.293 | 66.482 | 7.084 | 2.776 | 7.630 | 4.059 | 3.724 | 39.930 | 28.773 | 15.354 | 24.172 |
| $N_{\max} = 16$ | 8.698 | 108.335 | 35.467 | 9.008 | 2.215 | 5.147 | 5.808 | 4.475 | 40.162 | 25.936 | 10.165 | 23.220 |
| $\tau_{\text{conf}} = 0.25$ | 6.477 | 86.393 | 63.842 | 8.793 | 2.572 | 6.939 | 3.625 | 3.358 | 34.671 | 26.203 | 16.048 | 23.538 |
| $\tau_{\text{conf}} = 0.50$ | 6.788 | 83.293 | 66.482 | 7.084 | 2.776 | 7.630 | 4.059 | 3.724 | 39.930 | 28.773 | 15.354 | 24.172 |
| $\tau_{\text{conf}} = 0.75$ | 5.643 | 93.762 | 58.467 | 5.895 | 2.740 | 7.064 | 10.398 | 3.983 | 36.902 | 31.665 | 13.577 | 24.554 |
| $\tau_{\text{palx}} = 10$ | 7.114 | 89.507 | 48.887 | 10.330 | 5.128 | 8.806 | 3.332 | 4.153 | 35.618 | 33.407 | 21.271 | 24.323 |
| $\tau_{\text{palx}} = 15$ | 6.788 | 83.293 | 66.482 | 7.084 | 2.776 | 7.630 | 4.059 | 3.724 | 39.930 | 28.773 | 15.354 | 24.172 |
| $\tau_{\text{palx}} = 20$ | 7.290 | 88.862 | 51.228 | 9.608 | 1.491 | 7.482 | 5.296 | 4.511 | 41.000 | 29.624 | 17.680 | 23.830 |

| Waymo | | | | | | | | | |
|---|---|---|---|---|---|---|---|---|---|
| **Setting** | 163453191 | 183829460 | 315615587 | 346181117 | 371159869 | 405841035 | 460417311 | 520018670 | 610454533 | Avg. |
| $\tau_{\text{flow}} = 0.4$ | 1.289 | 0.139 | 0.363 | 3.096 | 1.893 | 1.081 | 2.160 | 3.933 | 0.121 | 1.564 |
| $\tau_{\text{flow}} = 0.7$ | 1.346 | 0.139 | 0.403 | 3.096 | 2.003 | 1.081 | 2.160 | 3.933 | 0.121 | 1.587 |
| $\tau_{\text{flow}} = 1.0$ | 1.594 | 0.157 | 0.384 | 3.096 | 1.903 | 1.081 | 2.160 | 3.934 | 0.121 | 1.603 |
| $r_{\text{turn}} = 3$ | 1.346 | 0.139 | 0.363 | 2.943 | 2.126 | 1.056 | 2.160 | 4.527 | 0.121 | 1.642 |
| $r_{\text{turn}} = 5$ | 1.346 | 0.139 | 0.403 | 3.096 | 2.003 | 1.081 | 2.160 | 3.933 | 0.121 | 1.587 |
| $r_{\text{turn}} = 7$ | 1.346 | 0.139 | 0.363 | 3.096 | 1.903 | 1.081 | 2.160 | 3.602 | 0.121 | 1.535 |
| $r_{\text{static}} = 0.5$ | 2.430 | 0.170 | 0.363 | 3.096 | 1.021 | 0.970 | 2.160 | 3.598 | 0.121 | 1.548 |
| $r_{\text{static}} = 0.6$ | 1.346 | 0.139 | 0.403 | 3.096 | 2.003 | 1.081 | 2.160 | 3.933 | 0.121 | 1.587 |
| $r_{\text{static}} = 0.7$ | 1.110 | 0.139 | 0.363 | 3.096 | 1.903 | 1.081 | 2.160 | 3.933 | 0.121 | 1.545 |
| $N_{\max} = 8$ | 1.292 | 0.139 | 0.394 | 3.116 | 2.071 | 1.102 | 2.073 | 4.048 | 0.117 | 1.595 |
| $N_{\max} = 12$ | 1.346 | 0.139 | 0.403 | 3.096 | 2.003 | 1.081 | 2.160 | 3.933 | 0.121 | 1.587 |
| $N_{\max} = 16$ | 1.306 | 0.139 | 0.399 | 3.111 | 2.087 | 1.052 | 2.028 | 3.974 | 0.155 | 1.584 |
| $\tau_{\text{conf}} = 0.25$ | 1.406 | 0.139 | 0.410 | 3.438 | 2.261 | 1.077 | 2.721 | 3.610 | 0.116 | 1.686 |
| $\tau_{\text{conf}} = 0.50$ | 1.346 | 0.139 | 0.403 | 3.096 | 2.003 | 1.081 | 2.160 | 3.933 | 0.121 | 1.587 |
| $\tau_{\text{conf}} = 0.75$ | 1.247 | 0.139 | 0.380 | 3.443 | 1.748 | 1.135 | 2.290 | 3.290 | 0.125 | 1.533 |
| $\tau_{\text{palx}} = 10$ | 1.161 | 0.196 | 0.384 | 3.330 | 2.360 | 1.050 | 2.347 | 3.911 | 0.123 | 1.651 |
| $\tau_{\text{palx}} = 15$ | 1.346 | 0.139 | 0.403 | 3.096 | 2.003 | 1.081 | 2.160 | 3.933 | 0.121 | 1.587 |
| $\tau_{\text{palx}} = 20$ | 1.416 | 0.177 | 0.387 | 3.138 | 1.929 | 0.981 | 2.201 | 3.702 | 0.189 | 1.569 |

*Table 8.* Runtime comparison on KITTI and Waymo. Results are reported in seconds per frame.

| KITTI | | | | | | | | | | | | |
|---|---|---|---|---|---|---|---|---|---|---|---|---|
| **Method** | 00 | 01 | 02 | 03 | 04 | 05 | 06 | 07 | 08 | 09 | 10 | Avg. |
| DROID-SLAM | 0.143 | 0.136 | 0.144 | 0.172 | 0.275 | 0.135 | 0.172 | 0.139 | 0.143 | 0.143 | 0.157 | 0.160 |
| Ours | 0.117 | 0.129 | 0.112 | 0.117 | 0.203 | 0.110 | 0.208 | 0.290 | 0.109 | 0.138 | 0.107 | 0.149 |

| Waymo | | | | | | | | | |
|---|---|---|---|---|---|---|---|---|---|
| **Method** | 163453191 | 183829460 | 315615587 | 346181117 | 371159869 | 405841035 | 460417311 | 520018670 | 610454533 | Avg. |
| DROID-SLAM | 0.409 | 0.194 | 0.224 | 0.380 | 0.364 | 0.239 | 0.379 | 0.366 | 0.212 | 0.308 |
| Ours | 0.316 | 0.440 | 0.439 | 0.278 | 0.337 | 0.313 | 0.342 | 0.294 | 0.376 | 0.348 |

## C. Qualitative and Quantitative Results

In this section, we provide comprehensive qualitative and quantitative analyses to further validate the efficacy of VGGT-Motion. We first evaluate its versatility in non-driving scenarios using handheld datasets (Section C.1), followed by detailed visualizations of trajectories and reconstructed point maps across all evaluated benchmarks (Section C.2 & C.3).

### C.1. Evaluation on Handheld Sequences (TUM-Mono (Engel et al., 2016))

To further evaluate the versatility of VGGT-Motion beyond standard autonomous driving benchmarks, we conduct additional experiments on the TUM-Mono dataset. Unlike the structured, mostly planar motion in driving scenarios, this dataset consists of handheld sequences with aggressive rotations, rapid scale changes, and indoor-outdoor transitions. These characteristics pose significant challenges for monocular SLAM systems, particularly in maintaining scale and orientation under non-linear motion.

*Table 10.* Absolute Trajectory Error (ATE) RMSE (m) comparison on representative TUM-Mono sequences. **VGGT-Motion (Ours)** demonstrates superior robustness across all handheld sequences compared to submap-based baselines.

| Method | 17 | 18 | 26 | 35 | 38 | 39 | 45 | 46 | 47 | 48 | Avg. |
|---|---|---|---|---|---|---|---|---|---|---|---|
| VGGT-Long | 21.43 | 7.81 | 0.54 | 0.57 | 0.88 | 3.37 | 1.14 | 1.35 | 7.09 | 4.46 | 4.86 |
| VGGT-SLAM | 120.25 | 30.30 | 2.43 | 0.66 | 4.35 | 4.47 | 13.59 | 0.91 | 40.58 | 21.57 | 23.91 |
| **VGGT-Motion (Ours)** | **10.31** | **2.73** | **0.32** | **0.39** | **0.34** | **1.02** | **0.92** | **0.65** | **5.72** | **1.74** | **2.41** |

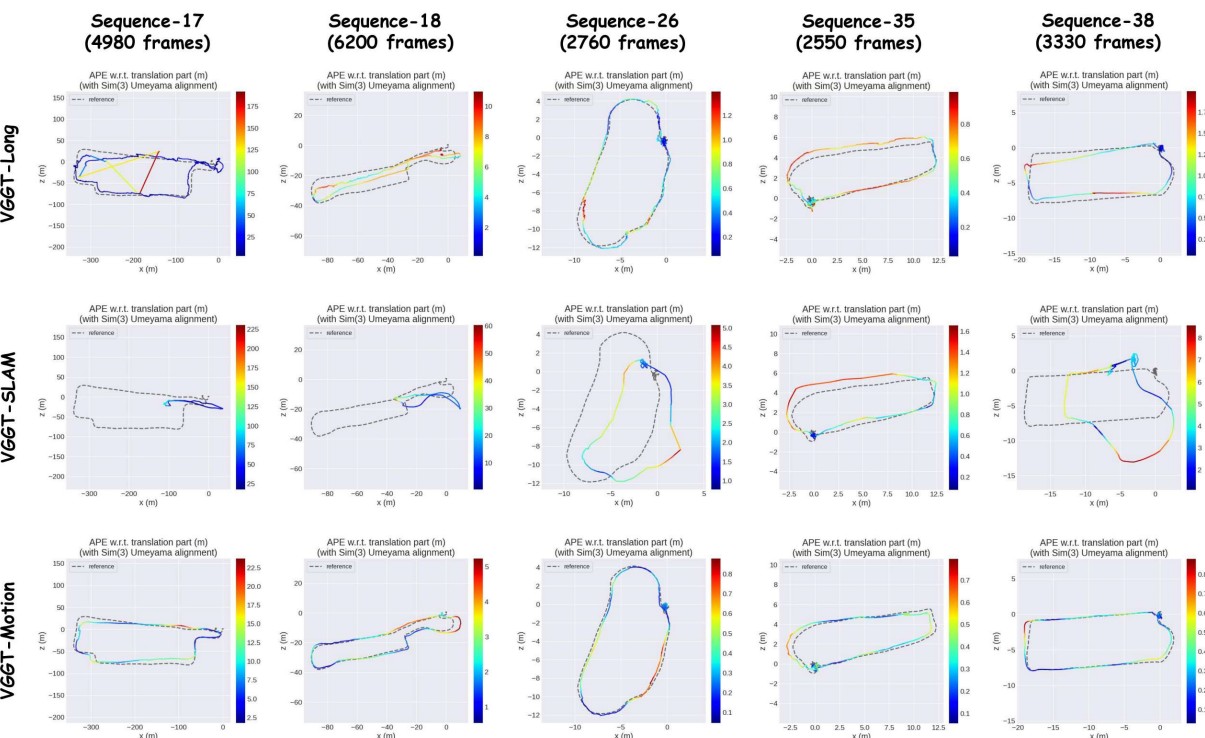

*Figure 11.* Trajectory visualizations on TUM-Mono sequences 17, 18, 26, 35, and 38. The color scale represents the Absolute Pose Error (APE) magnitude after $Sim(3)$ alignment.

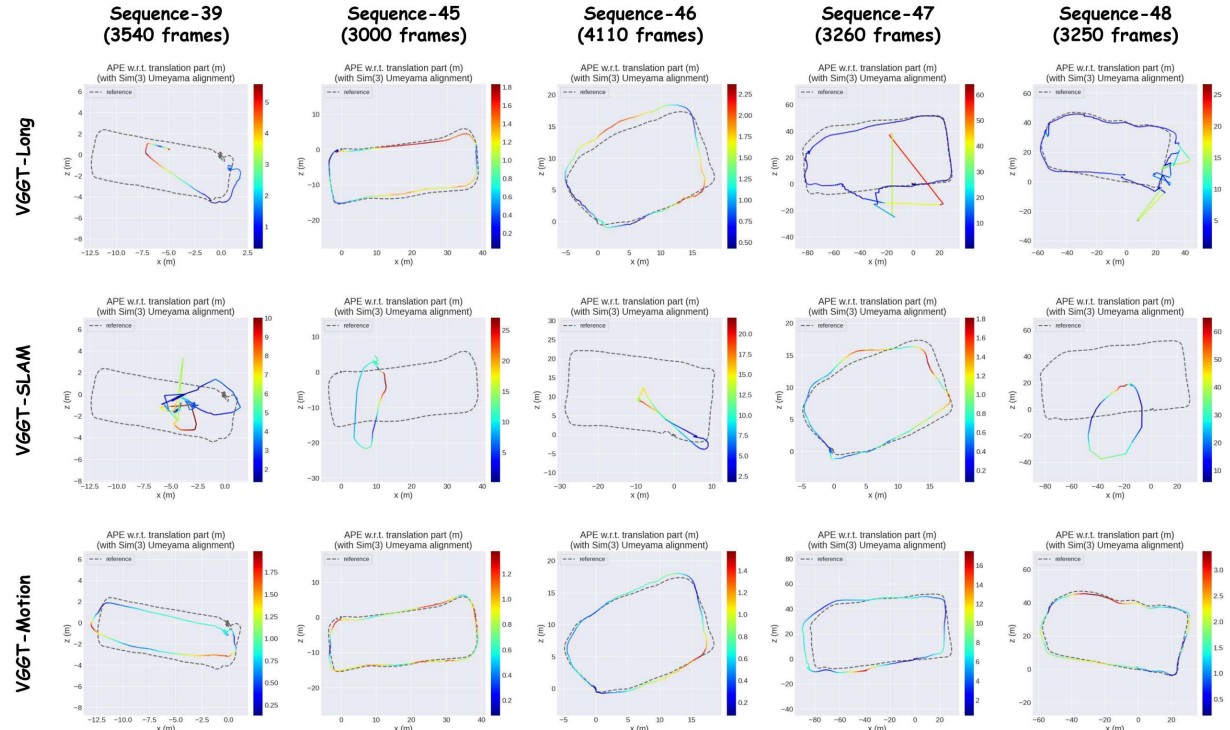

*Figure 12.* Additional trajectory visualizations on TUM-Mono sequences 39, 45, 46, 47, and 48. The color scale represents the Absolute Pose Error (APE) magnitude after $Sim(3)$ alignment.

**Analysis**

As shown in Table 10, our method achieves a significant performance leap over the baseline VGGT-SLAM and VGGT-Long. A key limitation of these baselines lies in its temporal slicing strategy, which partitions sequences into submaps based on fixed frame intervals. In handheld scenarios, this motion-agnostic partitioning frequently severs continuous turning maneuvers, disrupting the global self-attention context of the foundation model and leading to scale jumps at submap boundaries.

In contrast, our Motion-Aware Submap Construction leverages optical flow to perceive camera dynamics in real-time. By employing Turning Segment Encapsulation, we ensure that high-curvature maneuvers remain as indivisible geometric units within a single submap. This preserves structural coherence and provides a stable geometric baseline for monocular scale estimation. The qualitative results in Figure 11 and Figure 12 further confirm that our approach maintains a consistent global trajectory even during the complex head-turning motions prevalent in the TUM-Mono sequences, whereas baseline methods often exhibit catastrophic drift or tracking loss.

## C.2. Scalability in Trajectory Consistency

To demonstrate the scalability of VGGT-Motion, we provide detailed trajectory visualizations across sequences with varying durations, environments, and motion dynamics. Our system demonstrates consistent performance from short segments to ultra-long sequences containing tens of thousands of frames.

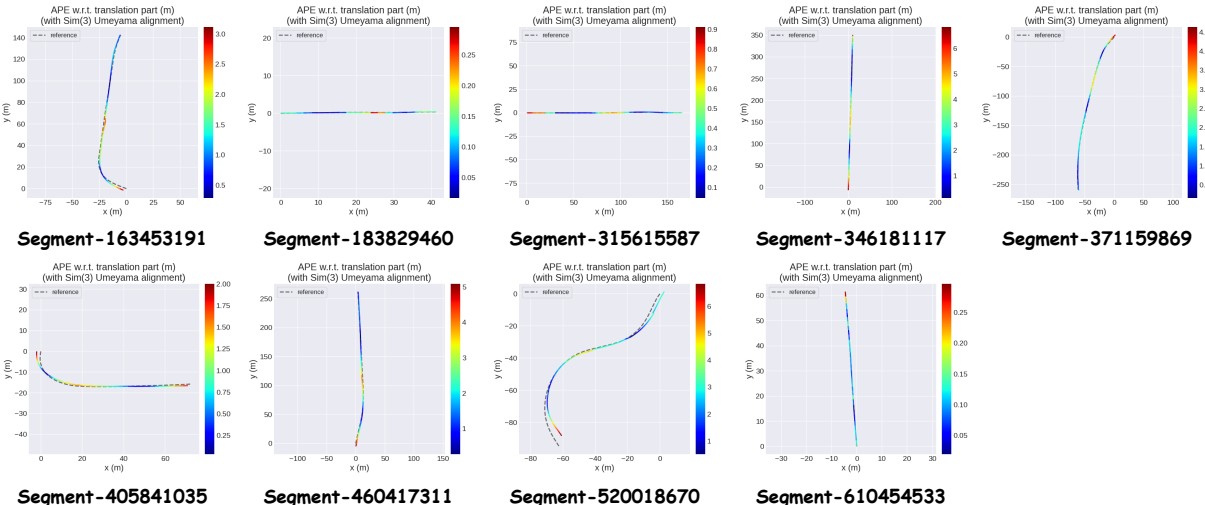

*Figure 13.* Trajectory visualizations on the Waymo Open dataset. Subplots show diverse segments, color denotes APE magnitude after $Sim(3)$ alignment.

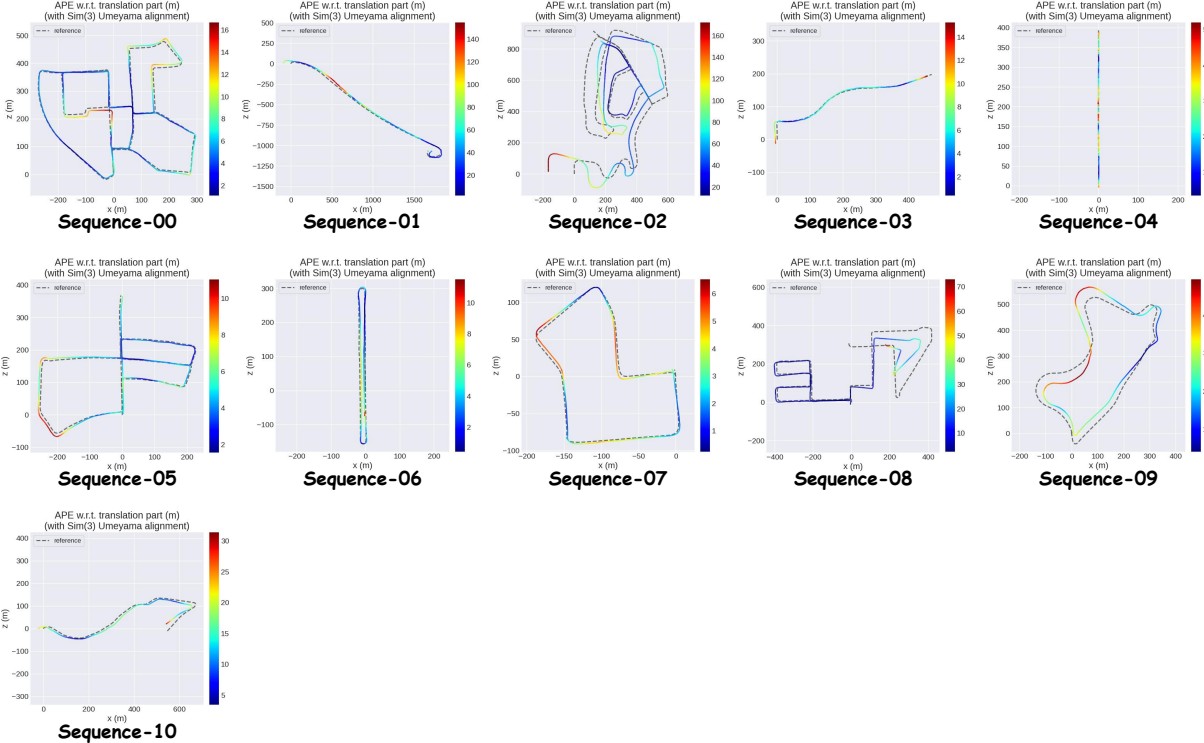

*Figure 14.* Trajectory visualizations on the KITTI benchmark. Subplots show diverse sequences, color denotes APE magnitude after $Sim(3)$ alignment.

*Table 11.* Absolute Trajectory Error (ATE) RMSE (m) comparison on long-range generalization benchmarks (4Seasons, Complex Urban, A2D2)

| Method | 4Seasons | | | | | Complex Urban | | | | | A2D2 | | |
|---|---|---|---|---|---|---|---|---|---|---|---|---|---|
| | Business-campus | Neighborhood | Office-loop | Old-town | Avg. | Sequence26 | Sequence27 | Sequence30 | Sequence38 | Avg. | Ingolstadt | Munich | Avg. |
| VGGT-Long | 133.13 | 120.62 | 256.71 | 610.53 | 280.25 | 346.78 | 326.10 | 707.90 | 521.59 | 475.59 | 176.65 | 189.21 | 182.93 |
| VGGT-SLAM | 33.61 | 142.57 | 299.38 | 196.60 | 168.04 | 343.64 | 316.83 | 635.51 | 482.56 | 444.63 | 192.39 | 193.20 | 192.80 |
| **VGGT-Motion(Ours)** | **4.66** | **6.51** | **14.44** | **23.27** | **12.22** | **17.40** | **24.80** | **53.21** | **46.53** | **35.48** | **33.97** | **25.64** | **29.80** |

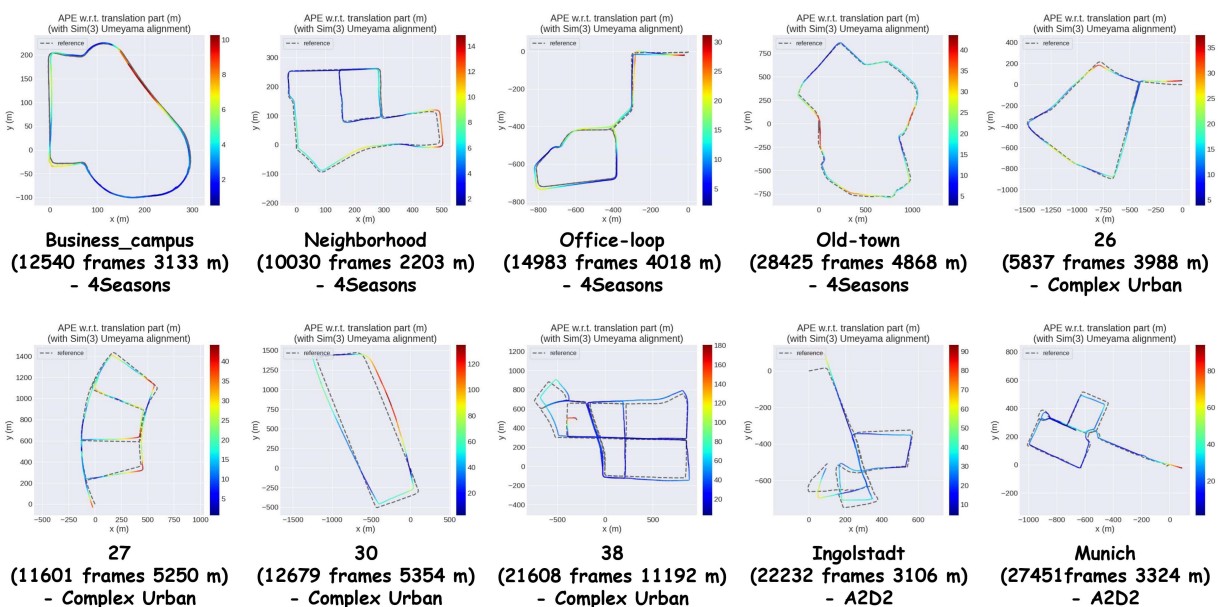

*Figure 15.* Trajectory visualizations on long-range generalization benchmarks (4Seasons, Complex Urban, A2D2). Subplots show diverse segments, color denotes APE magnitude after $Sim(3)$ alignment.

## Analysis

**Precision in Short, Loop-Free Segments (Waymo):** For short segments such as those in the Waymo Open Dataset (Figure 13), there are no loop closure opportunities to correct errors. In these cases, trajectory fidelity relies entirely on the accuracy of sequential submap alignment. Our Context-Balanced Anchor (CBA) strategy (Section 3.2) ensures that the overlapping frames used for registration possess optimal temporal "context." By mitigating the receptive field bias inherent in Transformer architectures, CBA provides sub-meter alignment precision, ensuring a seamless global trajectory even in the absence of loop constraints.

**Consistency in Moderate Sequences (KITTI):** In moderate-length sequences (Figure 14), the system balances local geometric fidelity with global efficiency. The $O(N)$ linear complexity of our submap-level pose graph optimization allows the system to scale gracefully, maintaining offline end-to-end runtime and long-range consistency without the memory bottlenecks associated with direct inference on long video streams.

**Suppression of Drift in Ultra-Long Sequences (4Seasons, Complex Urban, A2D2):** In sequences with tens of thousands of frames (Figure 15), the accumulation of "zero-motion drift" is the primary cause of failure. Such datasets often capture periods where the vehicle is stationary at traffic lights. While learned priors often "hallucinate" small displacements from sensor noise and environmental disturbances during these intervals, our Static Redundancy Pruning (Section 3.1) explicitly filters these redundant frames. By employing Parallax-based Keyframe Selection (Section 3.1), we prune these motion-redundant frames. This ensures a high-SNR geometric flow for scale consistency while significantly reducing the quadratic $O(N^2)$ computational overhead inherent in the foundation model's self-attention mechanism.

### C.3. High-Fidelity Geometric Reconstruction

Beyond global trajectory consistency, we evaluate the fidelity of local scene geometry through dense point cloud reconstructions across diverse environments.

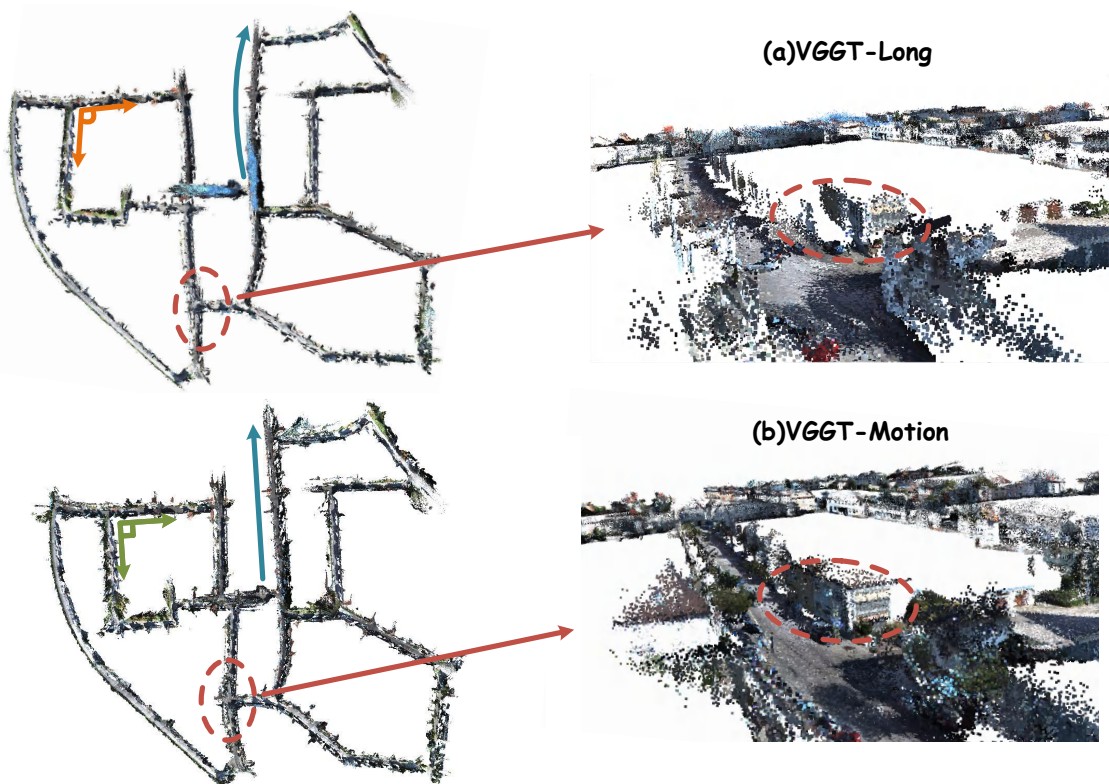

*Figure 16.* Point cloud reconstruction on KITTI sequence-00.

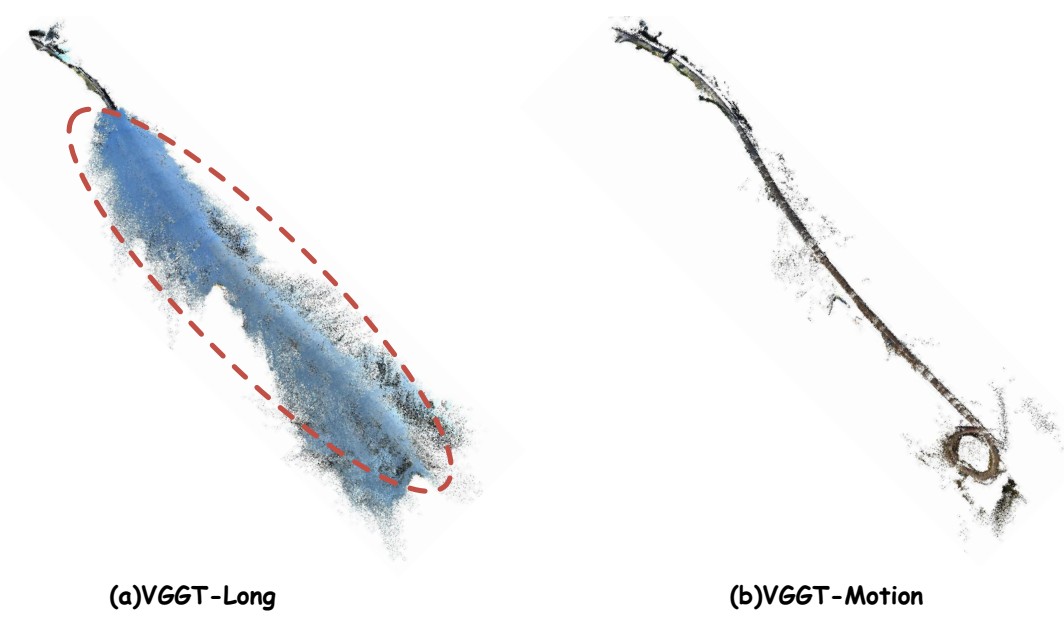

*Figure 17.* Point cloud reconstruction on KITTI sequence-01.

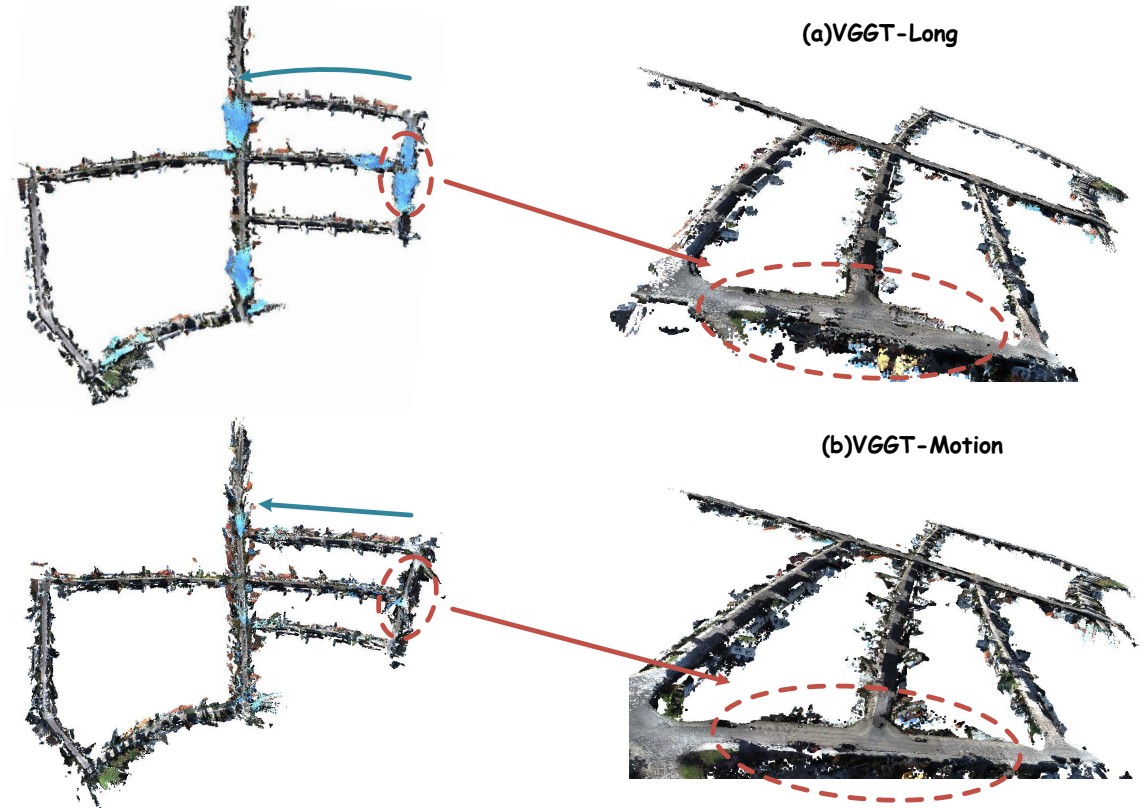

*Figure 18.* Point cloud reconstruction on the KITTI-sequence-05.

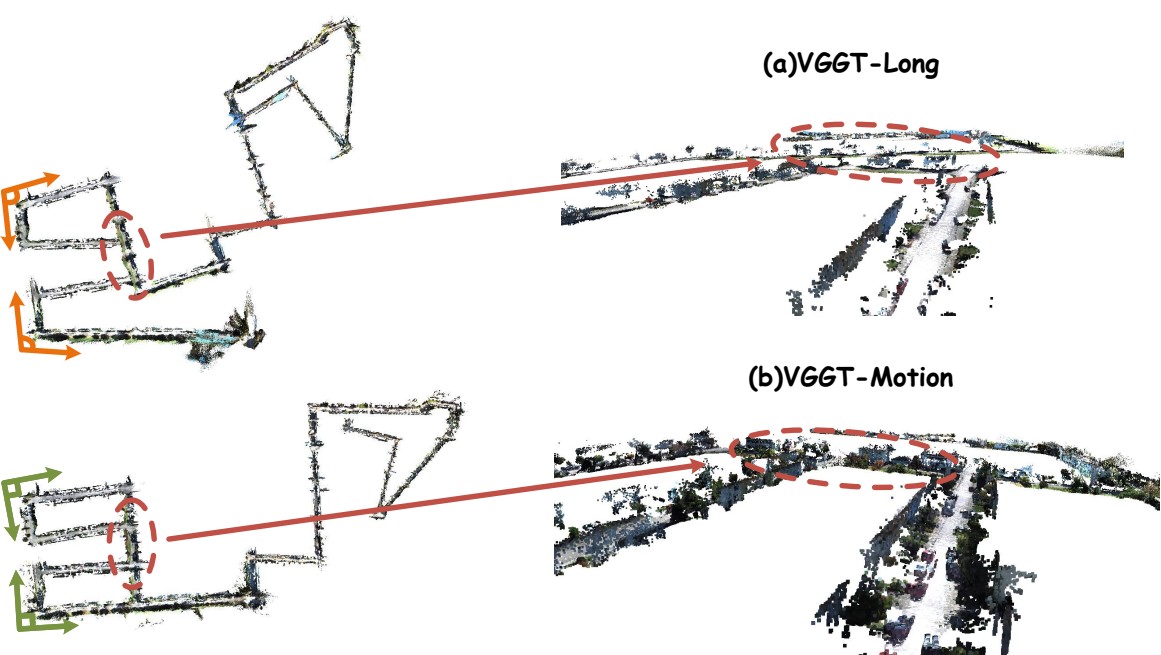

*Figure 19.* Point cloud reconstruction on the KITTI-sequence-08.

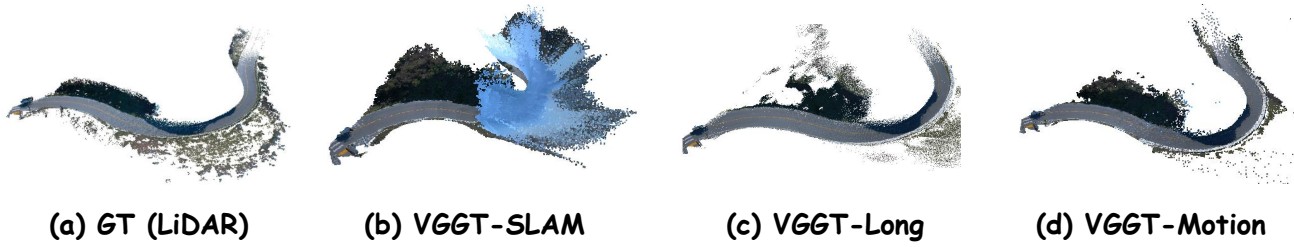

**(a) GT (LiDAR)**     **(b) VGGT-SLAM**     **(c) VGGT-Long**     **(d) VGGT-Motion**

*Figure 20.* Point cloud reconstruction on the Waymo-Segment-520018670.

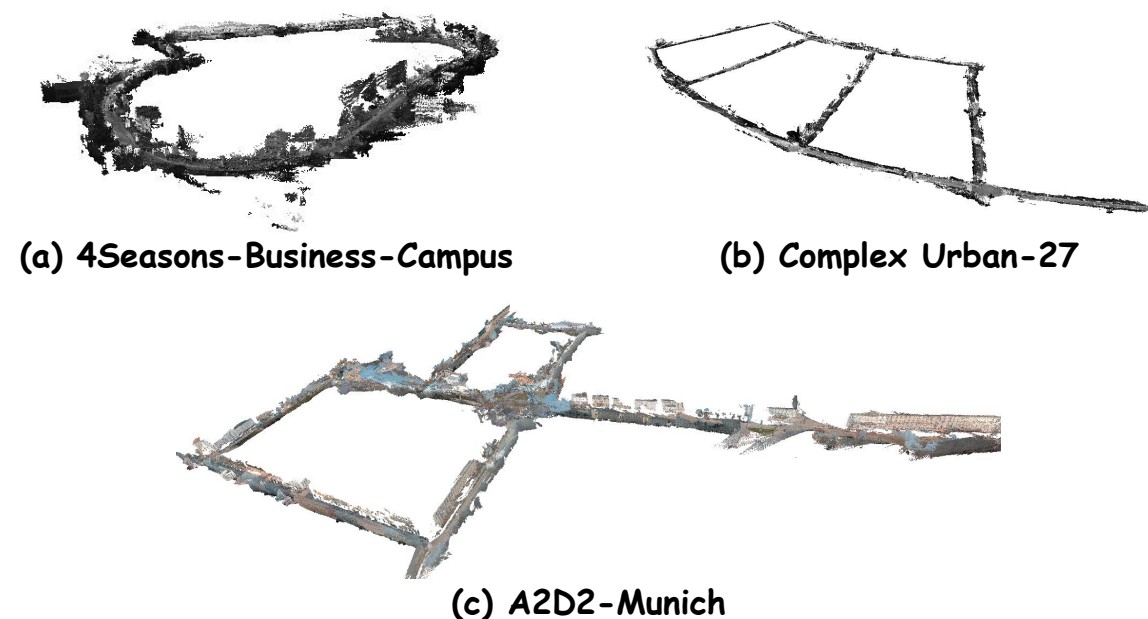

**(a) 4Seasons-Business-Campus**     **(b) Complex Urban-27**

**(c) A2D2-Munich**

*Figure 21.* Point cloud reconstruction on 4Seasons-Business Campus, Complex Urban-27, and A2D2-Munich.

**Analysis**

The Waymo segment (Figure 20) exemplifies Geometric Fragmentation (Section 1). While baselines (b, c) reconstruct coherent local maps, their rigid, motion-agnostic partitioning breaks turning maneuvers into disparate submaps. This disrupts the unified attention field inherent in Transformers, causing the observed scale jumps and global drift. In contrast, our Turning Segment Encapsulation (Section 3.1) preserves maneuvers as indivisible units, ensuring trajectory consistency despite complex dynamics.

Figure 17 and 18 highlight our suppression of sky-induced registration errors. Baselines often treat infinite-depth sky as finite, introducing systematic offsets during alignment. By masking semantic sky regions via $\Omega_{valid}$ (Section 3.2), our method achieves cleaner alignment using a 50% confidence threshold, compared with VGGT-Long using a 75% threshold. This proves that resolving geometric ambiguity at the source is more effective than aggressive post-inference filtering.

In long-range sequences (Figures 16, 18, 19, 21), our system effectively manages Spatio-temporal Redundancy (Section 1) through a dual-pronged pruning strategy. During stationary periods (e.g., traffic lights), Static Redundancy Pruning (Section 3.1) suppresses noise-driven hallucinated drift by filtering frames to lock the metric state. Simultaneously, during active driving, Parallax-based Keyframe Selection (Section 3.1) prunes motion-redundant frames that offer negligible geometric gain but cause semantic feature saturation. This synergy ensures a high-SNR geometric flow for scale consistency—preventing "ghosting" artifacts on structures like lamp posts—while significantly mitigating the quadratic $O(N^2)$ computational overhead of the transformer backend over kilometer-scale traversals.

