# OpenReview forum: "VGGT-Motion: Motion-Aware Calibration-Free Monocular SLAM for Long-Range Consistency"
_ICML.cc/2026/Conference — ICML 2026 spotlight_

### Official Review · Reviewer_eUYQ · 2026-03-12

**Soundness:** 3
**Presentation:** 3
**Significance:** 2
**Originality:** 2
**Overall Recommendation:** 4
**Confidence:** 4

**Summary:**

This paper proposes VGGT-Motion, a calibration-free monocular SLAM system built on top of the VGGT 3D vision foundation model. The method aims to improve long-range consistency and scalability for kilometer-scale trajectories. The key idea is a motion-aware submap construction strategy that classifies camera motion regimes using optical flow (static, linear, turning), prunes redundant frames, and encapsulates turning segments to preserve geometric context.

Each resulting submap is augmented with overlap and loop anchors and processed by the VGGT model to produce dense geometry predictions. Submaps are then aligned via an anchor-driven direct Sim(3) registration using pixel-wise dense correspondences derived from VGGT point maps, followed by a lightweight submap-level pose graph optimization.

Experiments on multiple datasets (KITTI, Waymo, 4Seasons, Complex Urban, A2D2) demonstrate large improvements over previous VGGT-based SLAM systems, reporting up to 85–95% reductions in trajectory error and substantial runtime improvements.

**Compliance With Llm Reviewing Policy:**

Affirmed.

**Key Questions For Authors:**

Please address the weakness above.

**Limitations:**

Yes.

**Strengths And Weaknesses:**

Strengths

1. Clear motivation and well-identified failure modes

The paper clearly identifies practical failure modes in foundation-model-based SLAM pipelines, including zero-motion hallucination during static intervals and geometric fragmentation when naive chunking cuts through turning motion. These issues are convincingly motivated.

2. Effective motion-aware segmentation design

The motion-aware partitioning strategy (optical-flow-based motion classification, parallax-based keyframe selection, and turning encapsulation) is simple yet well motivated. The design explicitly attempts to preserve geometric context for monocular scale estimation.

3. Efficient dense alignment formulation

The anchor-driven Sim(3) registration avoids feature matching by leveraging pixel-wise correspondences derived from VGGT point maps. This yields a conceptually clean alignment strategy with linear complexity and low overhead.

4. Strong empirical performance across datasets

The method demonstrates strong trajectory accuracy on several benchmarks, including kilometer-scale sequences and zero-shot datasets. The reported runtime improvements relative to VGGT-Long suggest that the proposed design scales well to longer sequences.



Weaknesses / Questions

1. Limited algorithmic novelty beyond segmentation

The main conceptual novelty appears to lie in the motion-aware submap segmentation strategy. Other components of the pipeline (dense Sim(3) alignment using foundation-model geometry and pose graph optimization) follow fairly standard designs in modern SLAM systems. It would be helpful to more clearly isolate the contribution of the segmentation strategy relative to the rest of the pipeline.

2. Hyperparameter sensitivity of motion-aware segmentation

The segmentation pipeline relies on several thresholds (e.g., optical-flow thresholds, parallax thresholds, maximum submap size). Although the paper states that these parameters are fixed across datasets, no sensitivity analysis is provided. Since motion classification directly determines the segmentation structure and the VGGT inference window, it would be important to understand how robust the method is across different motion regimes.

3. Robustness in dynamic or visually challenging scenes

The motion-aware segmentation relies on dense optical flow estimation and assumes mostly rigid scenes. In practice, optical flow estimation can degrade in low-texture environments, nighttime conditions, rain, or motion blur. In addition, dynamic objects such as vehicles and pedestrians are common in driving scenarios and may affect both motion estimation and geometry inference. The paper does not provide experiments evaluating robustness under such conditions.

4. Lack of analysis on VO-only performance

Since the main novelty lies in the motion-aware segmentation, it would be useful to better isolate its effect on the pipeline. In particular, the paper reports final SLAM results after pose graph optimization and loop closure. However, no results are provided for intermediate settings such as VO-only performance without loop closure, or pose graph BA. Such experiments would help clarify how much of the improvement originates from the motion-aware segmentation itself versus later global optimization stages.

Minor / Typo

- Fig.16 is incomplete.

---

> ### Author Rebuttal · Authors · 2026-03-29
>
> Dear Reviewer,
>
> We thank you for recognizing our work and for your professional review.
>
> ---
>
> > **Q1: Limited algorithmic novelty beyond segmentation**
>
> We appreciate your recognition the novelty of our submap segmentation. For submap alignment, we utilize dense Sim(3) alignment with our Context-Balanced Anchor (CBA) mechanism. CBA fundamentally overcomes the "Contextual Asymmetry" bottleneck in foundation models to eliminate scale drift, yielding massive performance gains over generic alignment (as shown in Fig. 6). Please refer to our response to **W1 of Reviewer ETfJ** for more discussion.
>
> > **Q2: Hyperparameter sensitivity of motion-aware segmentation**
>
> We agree that sensitivity analysis is crucial. We have conducted comprehensive tests on the 6 core hyperparameters controlling the segmentation pipeline on KITTI and Waymo benchmarks. The results are presented below. ***Default settings are bolded***.
>
> | Parameter | Setting | KITTI (avg) | Waymo (avg) |
> | :--- | :---: | :---: | :---: |
> | $\\tau_{flow}$ | 0.4 | 24.201 | 1.564 |
> | | **0.7** | **24.172** | **1.587** |
> | | 1.0 | 24.179 | 1.603 |
> | $\\tau_{turn}$ | 3 | 24.478 | 1.642 |
> | | **5** | **24.172** | **1.587** |
> | | 7 | 24.985 | 1.535 |
> | $\\tau_{static}$ | 0.5 | 24.147 | 1.548 |
> | | **0.6** | **24.172** | **1.587** |
> | | 0.7 | 24.197 | 1.545 |
> | $N_{max}$ | 8 | 29.900 | 1.595 |
> | | **12** | **24.172** | **1.587** |
> | | 16 | 23.220 | 1.584 |
> | $\\tau_{conf}$ | 0.25 | 23.538 | 1.686 |
> | | **0.50** | **24.172** | **1.587** |
> | | 0.75 | 24.554 | 1.533 |
> | $\\tau_{palx}$ | 10 | 24.323 | 1.651 |
> | | **15** | **24.172** | **1.587** |
> | | 20 | 23.830 | 1.569 |
>
> Results conclusively prove our method's robustness without overfitting. Even when adjusted within extreme margins (e.g., $N_{max} \in \{8, 12, 16\}$, $\tau_{palx} \in \{10, 15, 20\}$), trajectory accuracy remains remarkably stable. The Average Trajectory Error (ATE) fluctuates minimally (1.53m-1.68m) on the Waymo dataset, and mostly stabilizes around 24m on the KITTI dataset. Overall, instead of catastrophic tracking failures or severe scale drift, the system only experiences a marginal drop in accuracy.
>
> > **Q3: Robustness in dynamic or visually challenging scenes**
>
> Thank you for your valuable comment. We clarify the details as follows:
>
> - **Low texture, motion blur & weather.**
> To ensure a fair and rigorous evaluation, we benchmarked our system on five standard driving datasets, consistent with top-tier SLAM works. These datasets typically use high-frequency global shutter cameras with strict de-blurring preprocessing, so severe motion blur is rare.
> Since outdoor roads naturally possess rich structural textures, we explicitly filter out the primary textureless region (the sky) using a lightweight semantic segmentation model (L.268-270, left column).
> In addition, tests on 4Seasons (sunny, overcast, rain, snow) prove strong resistance to weather-induced visual degradation.
>
> - **Dynamic objects.**
> Our architecture decouples motion state classification from dense geometric inference.
> Optical flow only calculates macroscopic statistics. Local dynamic objects do not disrupt the global gating logic unless they occlude most of the view.
> The robust foundation model handles actual geometric inference.
> Furthermore, our pluggable framework allows seamless upgrades—replacing VGGT with a model like VGGT4D [1] will naturally yield further leaps in high-dynamic environments.
> | Foundation Model | Waymo-460417311 |
> | :--- | :---: |
> | VGGT | 2.159 |
> | **VGGT4D** | **1.739** |
>
> [1] Hu, Yu, et al. "VGGT4D: Mining Motion Cues in Visual Geometry Transformers for 4D Scene Reconstruction."
>
> > **Q4: Lack of analysis on VO-only performance**
>
> Our Pose Graph Optimization (PGO) strictly comprises sequential adjacent constraints and loop closure constraints. Crucially, global Bundle Adjustment (BA) is triggered exclusively upon explicit loop detection; otherwise, the backend reduces to sequential matrix multiplication without global optimization.
>
> Consequently, for sequences naturally lacking physical loops (e.g., KITTI 01, 03, 04, and all Waymo sequences), the final SLAM results reported in our main table ***reflect pure Visual Odometry (VO) performance***. The sustained high trajectory accuracy and low scale drift on these loopless sequences strongly validate our core claim: the massive performance gains stem from the robust local geometric constraints provided by our motion-aware segmentation and CBA mechanisms, rather than backend optimization.
>
> To directly address your concerns, we provide an ablation study on sequence with actual physical loops (KITTI 07) by disabling loop closure detection and PGO (i.e., a pure VO setting). The ATE results (in meters) are as follows:
>
> | Method | Setting | 07 |
> | :--- | :--- | :---: |
> | **VGGT-Long** | w/o LC | 16.64 |
> | | w/ LC | **4.04** |
> | **Ours** | w/o LC | 9.58 |
> | | w/ LC | **3.72** |
>
> ---
> We hope that we have accurately addressed any concerns that you may have had.

---

> > ### Author Rebuttal · Reviewer_eUYQ · 2026-04-03
> >
> > Thanks for the rebuttal. All my concerns are well addressed.

---

> > > ### Author Response · Authors · 2026-04-03
> > >
> > > Thank you very much for your time and for acknowledging our rebuttal. We deeply appreciate your constructive feedback throughout the review process, which has significantly helped us improve the quality of our paper. We are very glad to hear that our responses have fully addressed your concerns.

---

### Official Review · Reviewer_FSiM · 2026-03-12

**Soundness:** 3
**Presentation:** 4
**Significance:** 3
**Originality:** 3
**Overall Recommendation:** 4
**Confidence:** 5

**Summary:**

A relative high-speed and computation efficient calibration-free monocular SLAM.
Innovative Submap Partition and Direct Sim(3) Registration help to reduce the drift during long-distance localization.

**Compliance With Llm Reviewing Policy:**

Affirmed.

**Key Questions For Authors:**

1: "In contrast, we derive dense, search-free geometric correspondences from VGGT point maps to impose robust Sim(3) constraints" How to get this "correspondences " as you mentioned "search-free geometric"

2: compare with calibrated SLAM, how is the processing speed difference?

**Limitations:**

yes

**Strengths And Weaknesses:**

Strength:
1: A relative high-speed and computation efficient calibration-free monocular SLAM.
2: Innovative Submap Partition and Direct Sim(3) Registration help to reduce the drift during long-distance localization.
3: Numerus experiments to validate proposed algorithm
4: Qualitive results show the direct comparison with existing methods.

Weakness:
1: "In contrast, we derive dense, search-free geometric correspondences from VGGT point maps to impose robust Sim(3) constraints" How to get this "correspondences " as you mentioned "search-free geometric" should be more clear.

---

> ### Author Rebuttal · Authors · 2026-03-29
>
> Dear Reviewer,
>
> We thank you for recognizing our work and for your valuable feedback.
>
> ---
>
> > **W1 & Q1: Clarification on "search-free geometric correspondences"**
>
> We sincerely thank the reviewer for pointing out that this core mechanism needs further clarification.
>
> In traditional SLAM pipelines, obtaining geometric correspondences typically requires a highly time-consuming two-step operation: first extracting local visual descriptors, then performing explicit nearest neighbor searches (such as using KD-Trees or brute-force matching) between different images, and finally applying RANSAC to remove outliers.
>
> In contrast, our "search-free" mechanism is built directly upon the inherent network characteristics of the VGGT foundation model.
> Through deep global self-attention mechanisms, the Transformer implicitly completes multi-view geometric inference during the forward pass.
> VGGT does not output 2D features that require subsequent brute-force matching; instead, it directly regresses dense 3D point maps of the input images within a unified, shared 3D coordinate system.
>
> Because the network has already unified the 3D geometric structures of all pixels into this common reference frame, pixels pointing to the same physical 3D point are naturally aligned in the output space.
> Therefore, we only need to directly read these pre-aligned 3D coordinates (and filter out unreliable regions using the network's confidence scores) to establish dense 3D-3D correspondences.
> This entire process completely bypasses the explicit search phase of traditional descriptors, which is exactly what we mean by "search-free geometric correspondences".
>
> > **Q2: Processing speed difference compared with calibrated SLAM**
>
> To ensure a rigorous and fair evaluation, we specifically chose DROID-SLAM [1] as our baseline rather than sparse feature-based methods like ORB-SLAM2 [2]. Because DROID-SLAM also performs dense geometric mapping, the computational loads are much more comparable.
>
> Despite the extremely difficult task of performing zero-shot monocular SLAM with completely unknown camera intrinsics (calibration-free) while relying on a massive foundation model backend, our method remains highly competitive in processing speed. We conducted comprehensive evaluations on an NVIDIA RTX 3090 GPU, and in certain long sequence scenarios, our speed actually surpasses that of the calibrated DROID-SLAM.
>
> On short sequences (Waymo dataset), the processing speeds of both are basically identical.
> Our method takes approximately 0.348 s/frame (averaging ~21 seconds per sequence), while DROID-SLAM takes 0.308 s/frame (averaging ~20 seconds per sequence).
>
> On medium-to-long sequences (KITTI dataset), as the sequence length increases, our "motion-aware pruning strategy" demonstrates a significant efficiency advantage.
> Our processing speed reaches 0.149 s/frame (averaging ~92 seconds per sequence), which is actually faster than DROID-SLAM's 0.160 s/frame (averaging ~103 seconds per sequence)).
>
> A more critical difference lies in memory scalability on ultra-long trajectories.
> When evaluating real-world ultra-long sequences (e.g., KAIST, A2D2, 4Seasons), we found that DROID-SLAM's architecture requires pre-buffering massive amounts of GPU memory to maintain its Global Bundle Adjustment state graph.
> This leads to frequent Out-of-Memory (OOM) crashes on these long sequences.
>
> In stark contrast, our VGGT-Motion utilizes a topology-aware submap encapsulation strategy that explicitly limits context memory consumption, allowing the system to efficiently process kilometer-level trajectories without any OOM issues.
> In summary, our system not only matches state-of-the-art dense calibrated SLAM in processing speed but also fundamentally breaks the memory bottlenecks associated with processing long sequences.
>
> [1] Teed, Zachary, et al. "Droid-SLAM: Deep Visual Slam for Monocular, Stereo, and RGB-D Cameras."
> [2] Mur-Artal, Raul, et al. "ORB-SLAM2: An Open-Source Slam System for Monocular, Stereo, and RGB-D Cameras."
>
> ---
>
> We hope that we have accurately addressed any concerns that you may have had.

---

> > ### Author Rebuttal · Reviewer_FSiM · 2026-04-02
> >
> > Thanks for you response.
> >
> > I have some further questions:
> >
> > **Threshold Sensitivity**
> >
> > The motion estimation relies on multiple thresholds ($\tau_{flow}, \tau_{static}, \tau_{turn}, \tau_{palx}, \tau_{conf}, N_{max}, N_{ovlp}$). While the paper states these are "fixed across benchmarks," it does not provide a sensitivity analysis. Given that optical flow magnitudes are resolution- and scene-dependent, this is a non-trivial concern for deployment in truly unconstrained environments.
> >
> >
> > **Dynamic Object Handling**
> >
> > The system currently assumes a mostly rigid scene. In highly dynamic environments crowded with moving objects (e.g., busy intersections), the geometric priors from the foundation model might be contaminated, potentially affecting mapping consistency. This limitation is acknowledged, but no quantitative analysis of degradation under dynamic foreground density is provided.
> >
> >
> > **Bounded Global Error Guarantee**
> >
> > Similar to traditional monocular SLAM, the system faces accumulated scale, rotation, and translation drift, particularly during extended intervals between loop closures. Relying solely on learned visual priors is often insufficient to guarantee bounded global error; high-precision metric accuracy in large-scale environments still necessitates fusion with auxiliary sensors (e.g., IMU) to provide robust absolute constraints. This is a fundamental limitation inherent to the monocular modality, but it constrains the applicability of the system in safety-critical deployments.
> >
> >
> > **Comparison with VGGT-SLAM (SL(4))**
> >
> > The paper compares against VGGT-SLAM but does not deeply engage with its core theoretical argument — that projective SL(4) alignment is more principled than Sim(3) for uncalibrated cameras.NeurIPS Poster VGGT-SLAM: Dense RGB SLAM Optimized on the SL(4) ManifoldMore details about NeurIPS Poster VGGT-SLAM: Dense RGB SLAM Optimized on the SL(4) Manifold A more substantive discussion of why Sim(3) is sufficient (or its limitations relative to SL(4)) would strengthen the theoretical grounding.
> >
> >
> > **Loop Closure Detection Dependency**
> > The loop closure mechanism relies on SALAD-based retrieval, whose failure cases (e.g., perceptual aliasing, severe appearance change across seasons) are not analyzed. The 4Seasons dataset is explicitly multi-weather, making this a relevant concern.

---

> > > ### Author Response · Authors · 2026-04-02
> > >
> > > Dear Reviewer,
> > >
> > > We thank you for your valuable feedback.
> > >
> > > ---
> > >
> > > > **Threshold Sensitivity**
> > >
> > > We fully agree that sensitivity analysis is crucial. We have conducted additional evaluations on the 6 core thresholds across both the KITTI and Waymo benchmarks. For the detailed quantitative results, please kindly refer to our response to **Reviewer eUYQ (Q2)**.
> > >
> > > > **Dynamic Object Handling**
> > >
> > > To better understand our system's behavior in highly dynamic scenes (e.g., busy intersections), we evaluated our framework on Waymo's complex traffic sequences. Since our framework is designed to be modular, replacing the foundation model (VGGT) with **VGGT4D**[1]—a model specifically tailored for dynamic environments—yields quantitative improvements without requiring changes to our core SLAM logic, as shown below:
> > > | Foundation Model | segment-460417311 | segment-610454533 |
> > > | :--- | :---: | :---: |
> > > | VGGT | 2.159 | 0.121 |
> > > | **VGGT4D** | **1.739** | **0.115** |
> > >
> > > > **Bounded Global Error Guarantee**
> > >
> > > We deeply appreciate this insightful observation. We acknowledge that the inability to strictly bound global scale and metric drift over extended loop-less intervals is an inherent limitation of pure monocular setups. While our visual priors help mitigate this, we completely agree that safety-critical deployments would necessitate auxiliary sensors. Exploring a tightly coupled Visual-Inertial Odometry (VIO) framework to integrate IMU kinematics is a very important direction for our future work.
> > >
> > > > **Comparison with VGGT-SLAM (SL(4))**
> > >
> > > Thank you for raising this interesting theoretical point. During our early development, we actively experimented with optimizing on the SL(4) manifold. However, we observed that outdoor autonomous driving environments often introduce higher uncertainty (e.g., the infinite depth of the sky, complex dynamic objects, and distant backgrounds) compared to the indoor scenes where VGGT-SLAM was primarily validated.
> > >
> > > These outdoor uncertainties can introduce noise into the foundation model's depth predictions. Because the projective SL(4) manifold includes additional degrees of freedom—namely shear, stretch, and perspective—we found that optimizing on SL(4) in these scenarios sometimes overfitted the depth errors, leading to geometric distortions (such as visibly tilted roadside buildings). As noted by the VGGT-SLAM authors, this is a known vulnerability of projective manifolds. This theoretical limitation is further corroborated by our empirical findings; as shown in Table 1 of our manuscript, directly applying VGGT-SLAM to the outdoor KITTI dataset results in significant performance degradation.
> > >
> > > Consequently, we opted for the Sim(3) manifold. By preserving rigid angles and parallel structures, Sim(3) serves as a geometric regularizer, helping to prevent the system from absorbing depth noise into structural distortions in unconstrained outdoor scenes.
> > >
> > > > **Loop Closure Detection Dependency**
> > >
> > > We are grateful to the reviewer for highlighting these critical challenges in visual place recognition.
> > >
> > > **Regarding Perceptual Aliasing (False Positives)**: We do not blindly trust SALAD's retrieval. To prevent false loops from corrupting the Pose Graph, we employ strict geometric verification as a fail-safe. While calculating the transformation matrix between candidate submaps, we explicitly compute the geometric inlier ratio. If this ratio falls below our predefined threshold, the loop candidate is strictly rejected as a false positive.
> > >
> > > **Regarding Severe Appearance Changes**: This is a very valid concern. To better demonstrate our system's performance across different weather and appearance variations, we have included extensive evaluation results on the challenging 4Seasons dataset in the appendix (please see **Figure 15**), supplementing **Table 3** in the manuscript. Specifically, our method maintains relatively stable ATEs across these sequences: **Business_campus (4.70m), Neighborhood (6.65m), Office-loop (14.91m), and Old-town (23.01m)**.
> > >
> > > We believe this stability comes from two main aspects: 1) As demonstrated in the original SALAD paper [2] (Figure 5 in Izquierdo et al.), its DINOv2-powered architecture inherently provides reliable retrievals across weather and day-night variations. 2) Even if loops are occasionally missed, our highly accurate VO frontend (via the CBA mechanism) ensures graceful degradation, preventing catastrophic drift, as demonstrated by our pure-VO ablation on KITTI:
> > > | Method | Setting | Sequence 06 | Sequence 07 |
> > > | :--- | :--- | :---: | :---: |
> > > | **Ours** | w/o Loop Closure | 12.14 | 9.58 |
> > > | | w/ Loop Closure | **4.06** | **3.72** |
> > >
> > > [1] Hu, Yu, et al. "VGGT4D: Mining Motion Cues in Visual Geometry Transformers for 4D Scene Reconstruction."
> > >
> > > [2] Izquierdo, Javier, et al. "Optimal Transport Aggregation for Visual Place Recognition”
> > >
> > > ---
> > >
> > > We sincerely hope that these additional clarifications and experiments address your valuable concerns.

---

### Official Review · Reviewer_ETfJ · 2026-03-15

**Soundness:** 3
**Presentation:** 3
**Significance:** 3
**Originality:** 2
**Overall Recommendation:** 5
**Confidence:** 4

**Summary:**

This paper presents **VGGT-Motion**, a calibration-free monocular SLAM system built on the VGGT foundation model, designed to maintain global consistency over kilometer-scale trajectories. The method introduces three main components:
1. A *motion-aware submap construction* module that uses optical flow to classify frames into static, turning, and linear regimes, enabling adaptive keyframe selection, static redundancy pruning and turning segment encapsulation.
2. An *anchor-driven direct Sim(3) registration strategy* to achieve alignment between submaps without costly feature matching.
3. A *lightweight submap-level pose graph optimization* with linear complexity.

Experiments on five datasets (KITTI, Waymo, 4Seasons, Complex Urban, A2D2) demonstrates 85-95% ATE reduction and 18-36x speedup over VGGT-Long on long sequence benchmarks, along with competitive performance on standard benchmarks.

**Compliance With Llm Reviewing Policy:**

Affirmed.

**Final Justification:**

The paper presents a well-motivated and effective system for adapting 3D foundation models to long-sequence SLAM, with strong empirical results. While individual components draw on established techniques, the system-level integration is novel and addresses clearly identified failure modes.

The rebuttal fully addressed my concerns regarding optical flow dependency, hyperparameter sensitivity, and missing comparisons with streaming approaches such as TTT3R.

I therefore raise my rating to 5 (Accept).

**Key Questions For Authors:**

**1. Which optical flow method is used, and what is its computational overhead?**

**2. How sensitive is the system to the choice of hyperparameters?**

You state these are fixed across benchmarks, but driving scenarios might vary considerably. Are the hyperparameters the same for the evaluation on TUM-Mono?

**3. What happens if the geometric verification does not converge? (l.227, right column)**

You might have a hole in the full trajectory then, so how is the pose graph estimation done and how could we get a final complete reconstruction that would be scale-aligned for all submaps? This is not clear in the current description of the method.

**4. Why does CUT3R encounter OOM? (Table 1)**

CUT3R is an online method that should not encounter OOM when running on 12 frames for each submap. Maybe this line was inverted with MASt3R-SLAM. At least, something is not clear in this table. And what does LC mean? Long context?

Furthermore, what does TL mean for an online method like CUT3R? You should still be able to decode a pointmap from the state at each time. Is it for the final scale alignment? Or something else? Could you please develop?

**5. Minor details.**

In Figure 2 for (a) submap partitioning: first and last static tokens should be kept according to l.191, which seems confusing regarding Figure 2. Could you please change that?

**Limitations:**

Yes, in the appendix.

**Strengths And Weaknesses:**

**Strengths**

- **S1:** Well-motivated problem decomposition.

The paper clearly identifies three concrete failure modes of naive foundation model-based SLAM (geometric fragmentation, zero-motion drift, contextual asymmetry) and proposes targeted solutions for each. The problem analysis in Section 1 is insightful, particularly the observation that contextual asymmetry at submap boundaries causes systematic alignment bias due to differing receptive fields in the Transformer.

- **S2:** Strong empirical results on long sequences.

The improvements on the zero-shot generalization benchmarks are compelling. Reducing ATE from ~170m with VGGT-SLAM to ~12m with the proposed method on 4Seasons, and from ~445m to ~35m on Complex Urban represents a significant change. The 18-36x speedup is also practically significant.

- **S3:** Comprehensive evaluation.

The paper evaluates on five different datasets, including both calibrated and uncalibrated baselines, reports both ATE and Translation Drift, and provides thorough ablation studies isolating each component’s attribution. The additional TUM-Mono evaluation in the appendix demonstrates generalization beyond driving scenarios.

- **S4:** Model-agnostic design.

The discussion in Section B.2 showing the framework operating with Pi3 and DA3 strengthens the contribution by demonstrating modularity.

**Weaknesses**

- **W1:** Limited novelty in individual components.

While the system-level integration is effective, each individual component draws heavily on well established techniques. The primary novelty lies in adapting these ideas to the foundation model context, but the paper reads more like a well-engineered system based on already existing techniques.

- **W2:** Sensitivity analysis of hyperparameters is missing.

The system relies on multiple thresholds that are reportedly fixed across benchmarks. However, there is no sensitivity analysis showing how performance degrades as these hyperparameters vary. Given the diversity of motion patterns across datasets, it is unclear whether the chosen values are robust or if they were tuned to work across these specific autonomous driving based benchmarks.

- **W3:** Optical flow dependency is not discussed.

The motion state estimation relies on dense optical flow computation, but the paper does not specify which optical flow method is used, its computational cost, nor its failure modes. In textureless or highly dynamic scenes, optical flow estimation itself can be unreliable. This introduces a potential circular dependency between VGGT and the optical flow method, that deserves discussion.

- **W4:** Missing comparisons with recent streaming / causal approaches.

The related work mentions Stream3R, StreamVGGT, and TTT3R as concurrent work adopting streaming architectures, but none are included in the experimental comparison. Since these represent alternative paradigms that might not perform as well as the proposed method, their omission weakens the comparative evaluation. A comparison might actually strengthen the contribution given that streaming methods might still be prone to drift or catastrophic forgetting as stated in the paper.

---

> ### Author Rebuttal · Authors · 2026-03-29
>
> Dear Reviewer,
>
> We thank you for your thorough and insightful feedback.
>
> ---
>
> > **W1: Limited novelty in individual components.**
>
> As emphasized in our Introduction, our work is not a mere patchwork of existing techniques. Rather, it represents a first-of-its-kind system-level architectural innovation tailored to address three inherent bottlenecks exposed when adapting Transformer-based 3D foundation models for large-scale continuous inference: ***Geometric Fragmentation, Spatio-Temporal Redundancy, and Contextual Asymmetry***. Current SLAM paradigms cannot directly resolve the profound architectural conflicts between the underlying mechanisms of foundation models and continuous spatial reasoning. Our design fundamentally overcomes these adaptation bottlenecks.
>
> > **W2 & Q2: Sensitivity analysis of hyperparameters.**
>
> We keep the hyperparameters strictly fixed across all five autonomous driving datasets. However, the TUM-Mono dataset features hand-held sequences spanning both indoor and outdoor scenes, introducing high-frequency, irregular 6-DoF human jitter. To this, we slightly relax the turn-detection threshold. Furthermore, we have conducted a comprehensive hyperparameter sensitivity analysis. Please kindly refer to our response to **Q2 of Reviewer eUYQ** for the details.
>
> > **W3 & Q1: Optical flow dependency, chosen method, and computational overhead.**
>
> We use the classic Farneback algorithm in OpenCV tool to demonstrate our method's high compatibility. The computational overhead is lightweight: it takes an average of 3.3 ms per frame (with 95% of frames processed under 10 ms) and introduces only 522 MB of additional GPU memory usage.
>
> Furthermore, our system design avoids the following failure modes and circular dependencies:
>
> - **Textureless scenes.** Outdoor textureless regions are predominantly the "sky." We filter out these areas using a lightweight semantic segmentation model (L.268-270, left column), thereby preventing optical flow collapse.
> - **Dynamic objects.** In our approach, optical flow is used solely for macroscopic image-level statistics (e.g., static pixel ratio $r_{static}$, average lateral flow $m_{turn}$) rather than pixel-level matching, ensuring robustness against local anomalies.
> - **No circular dependency.** Optical flow is used exclusively to partition submaps. Since trajectory prediction and geometric reconstruction are independently handled by the foundation model, no circular dependency exists between VGGT and the optical flow module.
>
> > **W4: Missing comparisons with recent streaming / causal approaches.**
>
> Thank you for the interesting suggestion. We compared StreamVGGT and TTT3R on KITTI and Waymo, and the quantitative results (ATE in meters) are presented below:
>
> |Methods|KITTI (avg)|Waymo(avg)|
> |:---|:---:|:---:|
> |**StreamVGGT**|/|/|
> |**TTT3R**|68.127|3.340|
> |**Ours**|**24.17**|**1.59**|
>
> Due to its severe memory overhead, StreamVGGT consistently triggers Out-of-Memory (OOM) errors around frame 70 on a 24GB RTX 3090 GPU, making it difficult to predict the full trajectory. Although TTT3R avoids this issue, this pure streaming method continuously updates its implicit states, inevitably suffering from catastrophic forgetting and non-linear error accumulation. Consequently, its accuracy is significantly lower than that of our method.
>
> > **Q3: What happens if the geometric verification does not converge?**
>
> We clarify that our full trajectory contains no holes. The geometric verification is specifically designed to reject false positive loop closures, rather than to restrict adjacent submaps. For adjacent submap alignment, if not converged, we select the solution with the **highest inlier ratio** within max iterations derived iteratively from Eq. 10. Because adjacent submaps possess strong spatiotemporal continuity and co-visibility, these inlier ratios significantly exceed the predefined threshold. Conversely, when SALAD occasionally proposes incorrect loop candidates, geometric verification effectively rejects them.
>
> > **Q4: Why does CUT3R encounter OOM?**
>
> Unlike our 12-frame submap design, extracting full trajectories in CUT3R requires backend global alignment. As confirmed in its GitHub Issue #10, this post-processing accumulates massive memory, consistently triggering OOM even on short 300-frame sequences, making evaluation on long-sequence benchmarks infeasible. In addition,
> - **LC** stands for "Loop Closure". We are sorry for not noting this in the table caption.
> - **TL** stands for "Tracking Lost", as noted in the caption of Table 1. For an online method like CUT3R, tracking lost occurs when its persistent state diverges or crashes upon encountering challenging frames. Once this happens, the system can no longer decode valid poses or pointmaps for the remainder of the sequence.
>
> > **Q5: Minor details.**
>
> We are sorry for this oversight and will correct it in the final version.
>
> ---
>
> We hope that we have accurately addressed any concerns that you may have had.

---

> > ### Author Rebuttal · Reviewer_ETfJ · 2026-04-03
> >
> > We thank the authors for the strong rebuttal. My concerns are fully addressed, hence I will raise my rating.

---

> > > ### Author Response · Authors · 2026-04-03
> > >
> > > Thank you very much for your time and for acknowledging our rebuttal. We deeply appreciate your constructive feedback throughout the review process, which has significantly helped us improve the quality of our paper. We are very glad to hear that our responses have fully addressed your concerns.

---

### Decision · Program_Chairs · 2026-04-30

**Decision:**

Accept (spotlight)

**Comment:**

All the reviewers recommend accepting this submission after the rebuttal. The results are impressive and the paper is well written. The additional experiments done for the rebuttal should be added to the main paper or an appendix.